# BIDIRECTIONALLY SELF-NORMALIZING NEURAL NETWORKS

## ABSTRACT

The problem of exploding and vanishing gradients has been a long-standing obstacle that hinders the effective training of neural networks. Despite various tricks and techniques that have been employed to alleviate the problem in practice, there still lacks satisfactory theories or provable solutions. In this paper, we address the problem from the perspective of high-dimensional probability theory. We provide a rigorous result that shows, under mild conditions, how the exploding/vanishing gradient problem disappears with high probability if the neural networks have sufficient width. Our main idea is to constrain both forward and backward signal propagation in a nonlinear neural network through a new class of activation functions, namely Gaussian-Poincaré normalized functions, and orthogonal weight matrices. Experiments on both synthetic and real-world data validate our theory and confirm its effectiveness on very deep neural networks when applied in practice.

## 1 INTRODUCTION

Neural networks have brought unprecedented performance in various artificial intelligence tasks (Graves et al., 2013; Krizhevsky et al., 2012; Silver et al., 2017). However, despite decades of research, training neural networks is still mostly guided by empirical observations and successful training often requires various heuristics and extensive hyperparameter tuning. It is therefore desirable to understand the cause of the difficulty in neural network training and to propose theoretically sound solutions.

A major difficulty is the gradient exploding/vanishing problem (Glorot & Bengio, 2010; Hochreiter, 1991; Pascanu et al., 2013; Philipp et al., 2018). That is, the norm of the gradient in each layer is either growing or shrinking at an exponential rate as the gradient signal is propagated from the top layer to bottom layer. For deep neural networks, this problem might cause numerical overflow and make the optimization problem intrinsically difficult, as the gradient in each layer has vastly different magnitude and therefore the optimization landscape becomes pathological. One might attempt to solve the problem by simply normalizing the gradient in each layer. Indeed, the adaptive gradient optimization methods (Duchi et al., 2011; Kingma & Ba, 2015; Tieleman & Hinton, 2012) implement this idea and have been widely used in practice. However, one might also wonder if there is a solution more intrinsic to deep neural networks, whose internal structure if well-exploited would lead to further advances.

To enable the trainability of deep neural networks, batch normalization (Ioffe & Szegedy, 2015) was proposed in recent years and achieved widespread empirical success. Batch normalization is a differentiable operation which normalizes its inputs based on mini-batch statistics and inserted between the linear and nonlinear layers. It is reported that batch normalization can accelerate neural network training significantly (Ioffe & Szegedy, 2015). However, batch normalization does not solve the gradient exploding/vanishing problem (Philipp et al., 2018). Indeed it is proved that batch normalization can actually worsen the problem (Yang et al., 2019). Besides, batch normalization requires separate training and testing phases and might be ineffective when the mini-batch size is small (Ioffe, 2017). The shortcomings of batch normalization motivate us to search for a more principled and generic approach to solve the gradient exploding/vanishing problem.

Alternatively, self-normalizing neural networks (Klambauer et al., 2017) and dynamical isometry theory (Pennington et al., 2017) were proposed to combat gradient exploding/vanishing problem. In self-normalizing neural networks, the output of each network unit is constrained to have zero mean and unit variance. Based on this motivation, a new activation function, scaled exponential linear unit (SELU), was devised. In dynamical isometry theory, all singular values of the input-output Jacobian matrix are constrained to be close to one at initialization. This amounts to initializing the functionality of a network to be close to an orthogonal matrix. While the two theories dispense batch normalization, it is shown that SELU still suffers from exploding/vanishing gradient problem and dynamical isometry restricts the functionality of the network to be close to linear (pseudo-linearity) (Philipp et al., 2018).

In this paper, we follow the above line of research to investigate neural network trainability. Our contributions are three-fold: First, we introduce bidirectionally self-normalizing neural network (BSNN) that consist of orthogonal weight matrices and a new class of activation functions which we call Gaussian-Poincaré normalized (GPN) functions. We show many common activation functions can be easily transformed into their respective GPN versions. Second, we rigorously prove that the gradient exploding/vanishing problem disappears with high probability in BSNNs if the width of each layer is sufficiently large. Third, with experiments on synthetic and real-world data, we confirm that BSNNs solve the gradient vanishing/exploding problem to large extent while maintaining nonlinear functionality.

## 2 THEORY

In this section, we introduce bidirectionally self-normalizing neural network (BSNN) formally and analyze its properties. All the proofs of our results are left to Appendix.

To simplify the analysis, we define a neural network in a restricted sense as follows:

**Definition 1** (**Neural Network**). *A neural network is a function from $\mathbb{R}^n$ to $\mathbb{R}^n$ composed of layer-wise operations for $l = 1, \ldots, L$ as*

$$\mathbf{h}^{(l)} = \mathbf{W}^{(l)}\mathbf{x}^{(l)}, \quad \mathbf{x}^{(l+1)} = \phi(\mathbf{h}^{(l)}), \tag{1}$$

*where $\mathbf{W}^{(l)} \in \mathbb{R}^{n \times n}$, $\phi : \mathbb{R} \to \mathbb{R}$ is a differentiable function applied element-wise to a vector, $\mathbf{x}^{(1)}$ is the input and $\mathbf{x}^{(L+1)}$ is the output.*

Under this definition, $\phi$ is called the activation function, $\{\mathbf{W}^{(l)}\}_{l=1}^{L}$ are called the parameters, $n$ is called the width and $L$ is called the depth. Superscript $(l)$ denotes the $l$-th layer of a neural network. The above formulation is similar to (Pennington et al., 2017) but we omit the bias term in (1) for simplicity as it plays no role in our analysis.

Let $E$ be the objective function of $\{\mathbf{W}^{(l)}\}_{l=1}^{L}$ and $\mathbf{D}^{(l)}$ be a diagonal matrix with diagonal elements $D_{ii}^{(l)} = \phi'(h_i^{(l)})$, where $\phi'$ denotes the derivative of $\phi$. Now, the error signal is back propagated via

$$\mathbf{d}^{(L)} = \mathbf{D}^{(L)}\frac{\partial E}{\partial \mathbf{x}^{(L+1)}}, \quad \mathbf{d}^{(l)} = \mathbf{D}^{(l)}(\mathbf{W}^{(l+1)})^T\mathbf{d}^{(l+1)}, \tag{2}$$

and the gradient of the weight matrix for layer $l$ can be computed as

$$\frac{\partial E}{\partial \mathbf{W}^{(l)}} = \mathbf{d}^{(l)}(\mathbf{x}^{(l)})^T. \tag{3}$$

To solve the gradient exploding/vanishing problem, we constrain the forward signal $\mathbf{x}^{(l)}$ and the backward signal $\mathbf{d}^{(l)}$ in order to constrain the norm of the gradient. This leads to the following definition and proposition.

**Definition 2** (**Bidirectional Self-Normalization**). *A neural network is bidirectionally self-normalizing if*

$$\|\mathbf{x}^{(1)}\|_2 = \|\mathbf{x}^{(2)}\|_2 = \ldots = \|\mathbf{x}^{(L)}\|_2, \tag{4}$$

$$\|\mathbf{d}^{(1)}\|_2 = \|\mathbf{d}^{(2)}\|_2 = \ldots = \|\mathbf{d}^{(L)}\|_2. \tag{5}$$

**Proposition 1.** *If a neural network is bidirectionally self-normalizing, then*

$$\left\|\frac{\partial E}{\partial \mathbf{W}^{(1)}}\right\|_F = \left\|\frac{\partial E}{\partial \mathbf{W}^{(2)}}\right\|_F = \ldots = \left\|\frac{\partial E}{\partial \mathbf{W}^{(L)}}\right\|_F. \tag{6}$$

In the rest of this section, we derive the conditions under which bidirectional self-normalization is achievable for a neural network.

## 2.1 Constraints on Weight Matrices

We constrain the weight matrices to be orthogonal since multiplication by an orthogonal matrix preserves the norm of a vector. For linear networks, this guarantees bidirectionally self-noramlizingnormalization and its further benefits are discussed in (Saxe et al., 2014). Even for nonlinear neural networks, orthogonal constraints are shown to improve the trainability with proper scaling (Mishkin & Matas, 2016; Pennington et al., 2017).

## 2.2 Constraints on Activation Functions

To achieve bidirectionally self-noramlizingnormalization for a nonlinear network, it is not enough only to constrain the weight matrices. We also need to constrain the activation function in such a way that both forward and backward signals are normalized. To this end, we propose the following constraint that captures the relationship between a function and its derivative.

**Definition 3** (**Gaussian-Poincaré Normalization**). *Function $\phi : \mathbb{R} \to \mathbb{R}$ is Gaussian-Poincaré normalized if it is differentiable and*

$$\mathbb{E}_{x \sim \mathcal{N}(0,1)}[\phi(x)^2] = \mathbb{E}_{x \sim \mathcal{N}(0,1)}[\phi'(x)^2] = 1. \tag{7}$$

The definition is inspired by the following theorem which shows the fundamental relationship between a function and its derivative under Gaussian measure.

**Theorem 1** (**Gaussian-Poincaré Inequality** (Bogachev, 1998)). *If function $\phi : \mathbb{R} \to \mathbb{R}$ is differentiable with bounded $\mathbb{E}_{x \sim \mathcal{N}(0,1)}[\phi(x)^2]$ and $\mathbb{E}_{x \sim \mathcal{N}(0,1)}[\phi'(x)^2]$, then*

$$\mathrm{Var}_{x \sim \mathcal{N}(0,1)}[\phi(x)] \leq \mathbb{E}_{x \sim \mathcal{N}(0,1)}[\phi'(x)^2]. \tag{8}$$

Note that there is an implicit assumption that the input is approximately Gaussian for a Gaussian-Poincaré normalized (GPN) function. Even though this is standard in the literature (Klambauer et al., 2017; Pennington et al., 2017; Schoenholz et al., 2017), we will rigorously prove that this assumption is valid when orthogonal weight matrices are used in equation 1. Next, we state a property of GPN functions.

**Proposition 2.** *Function $\phi : \mathbb{R} \to \mathbb{R}$ is Gaussian-Poincaré normalized and $\mathbb{E}_{x \sim \mathcal{N}(0,1)}[\phi(x)] = 0$ if and only if $\phi(x) = x$ or $\phi(x) = -x$.*

This result indicates that any nonlinear function with zero mean under Gaussian distribution (*e.g.*, Tanh and SELU) is not GPN. Now we show that a large class of activation functions can be converted into their respective GPN versions using an affine transformation.

**Proposition 3.** *For any differentiable function $\phi : \mathbb{R} \to \mathbb{R}$ with non-zero and bounded $\mathbb{E}_{x \sim \mathcal{N}(0,1)}[\phi(x)^2]$ and $\mathbb{E}_{x \sim \mathcal{N}(0,1)}[\phi'(x)^2]$, there exist two constants $a$ and $b$ such that $a\phi(x) + b$ is Gaussian-Poincaré normalized.*

To obtain $a$ and $b$, one can use numerical procedure to compute the values of $\mathbb{E}_{x \sim \mathcal{N}(0,1)}[\phi'(x)^2]$, $\mathbb{E}_{x \sim \mathcal{N}(0,1)}[\phi(x)^2]$ and $\mathbb{E}_{x \sim \mathcal{N}(0,1)}[\phi(x)]$ and then solve the quadratic equations

$$\mathbb{E}_{x \sim \mathcal{N}(0,1)}[a^2\phi'(x)^2] = 1, \tag{9}$$

$$\mathbb{E}_{x \sim \mathcal{N}(0,1)}[(a\phi(x) + b)^2] = 1. \tag{10}$$

We computed $a$ and $b$ (not unique) for several common activation functions with their default hyperparameters[1] and the results are listed in Table 1. Note that ReLU, LeakyReLU and SELU are not differentiable at $x = 0$ but they can be regarded as approximations of their smooth counterparts. We ignore such point and evaluate the integrals for $x \in (-\infty, 0) \cup (0, \infty)$.

With the orthogonal constraint on the weight matrices and the Gaussian-Poincaré normalization on the activation function, we prove that bidirectionally self-noramlizingnormalization is achievable with high probability under mild conditions in the next subsection.

---

[1] We use $\alpha = 0.01$ for LeakyReLU, $\alpha = 1$ for ELU and $\phi(x) = x/(1 + \exp(-1.702x))$ for GELU.

| | Tanh | ReLU | LeakyReLU | ELU | SELU | GELU |
|---|---|---|---|---|---|---|
| $a$ | 1.4674 | 1.4142 | 1.4141 | 1.2234 | 0.9660 | 1.4915 |
| $b$ | 0.3885 | 0.0000 | 0.0000 | 0.0742 | 0.2585 | $-0.9097$ |

Table 1: Constants for Gaussian-Poincaré normalization of activation functions.

## 2.3 NORM-PRESERVATION THEOREMS

The bidirectionally self-noramlizingnormalization may not be achievable precisely in general unless the neural network is a linear one. Therefore, we investigate the properties of neural networks in a probabilistic framework. The random matrix theory and the high-dimensional probability theory allow us to characterize the behaviors of a large class of neural networks by its mean behavior, which is significantly simpler to analyze. Therefore, we study neural networks of random weights whose properties may shed light on the trainability of neural networks in practice.

First, we need a probabilistic version of the vector norm constraint.

**Definition 4 (Thin-Shell Concentration).** *Random vector* $\mathbf{x} \in \mathbb{R}^n$ *is thin-shell concentrated if for any* $\epsilon > 0$

$$\mathbb{P}\left\{\left|\frac{1}{n}\|\mathbf{x}\|_2^2 - 1\right| \geq \epsilon\right\} \to 0 \tag{11}$$

*as* $n \to \infty$.

The definition is modified from the one in (Bobkov, 2003). Examples of thin-shell concentrated distributions include standard multivariate Gaussian and any distribution on the $n$-dimensional sphere of radius $\sqrt{n}$.

**Assumptions.** To prove the main results, *i.e.*, the norm-preservation theorems, we require the following assumptions:

1. *Random vector* $\mathbf{x} \in \mathbb{R}^n$ *is thin-shell concentrated.*
2. *Random orthogonal matrix* $\mathbf{W} = (\mathbf{w}_1, \mathbf{w}_2, ..., \mathbf{w}_n)^T$ *is uniformly distributed.*
3. *Function* $\phi : \mathbb{R} \to \mathbb{R}$ *is Gaussian-Poincaré normalized.*
4. *Function* $\phi$ *and its derivative are Lipschitz continuous.*

The above assumptions are not restrictive. For Assumption 1, one can always normalize the input vectors of a neural network. For Assumption 2, orthogonal constraint or its relaxation has already been employed in neural network training (Brock et al., 2017). Note, in Assumption 2, uniformly distributed means that $\mathbf{W}$ is distributed under Haar measure, which is the unique rotation invariant probability measure on orthogonal matrix group. We refer the reader to (Meckes, 2019) for details. Furthermore, all the activation functions or their smooth counterparts listed in Table 1 satisfy Assumptions 3 and 4.

With the above assumptions, we can prove the following norm-preservation theorems.

**Theorem 2 (Forward Norm-Preservation).** *Random vector*

$$(\phi(\mathbf{w}_1^T\mathbf{x}), \phi(\mathbf{w}_2^T\mathbf{x}), ..., \phi(\mathbf{w}_n^T\mathbf{x})) \tag{12}$$

*is thin-shell concentrated.*

This result shows the transformation (orthogonal matrix followed by the GPN activation function) can preserve the norm of its input with high probability. Since the output is thin-shell concentrated, it serves as the input for the next layer and so on. Hence, the forward pass can preserve the norm of its input in each layer along the forward path when $n$ is sufficiently large.

**Theorem 3 (Backward Norm-Preservation).** *Let* $\mathbf{D}$ *be the diagonal matrix whose diagonal elements are* $D_{ii} = \phi'(\mathbf{w}_i^T\mathbf{x})$ *and* $\mathbf{y} \in \mathbb{R}^n$ *be a fixed vector with bounded* $\|\mathbf{y}\|_\infty$. *Then for any* $\epsilon > 0$

$$\mathbb{P}\left\{\frac{1}{n}\left|\|\mathbf{D}\mathbf{y}\|_2^2 - \|\mathbf{y}\|_2^2\right| \geq \epsilon\right\} \to 0 \tag{13}$$

*as* $n \to \infty$.

This result shows that the multiplication by $\mathbf{D}$ preserves the norm of its input with high probability. Since orthogonal matrix $\mathbf{W}$ also preserves the norm of its input, when the gradient error signal is propagated backwards as in (2), the norm is preserved in each layer along the backward path when $n$ is sufficient large.

Hence, combining Theorems 2 and 3, we proved that bidirectionally self-noramlizingnormalization is achievable with high probability if the neural network is wide enough and the conditions in the Assumptions are satisfied. Then by Proposition 1, the gradient exploding/vanishing problem disappears with high probability.

**Sketch of the proofs.** The proofs of Theorems 2 and 3 are mainly based on a phenomenon in high-dimensional probability theory, concentration of measure. We refer the reader to (Vershynin, 2018) for an introduction to the subject. Briefly, it can be shown that for some high-dimensional probability distributions, most mass is concentrated around a certain range. For example, while most mass of a low-dimensional standard multivariate Gaussian distribution is concentrated around the center, most mass of a high-dimensional standard multivariate Gaussian distribution is concentrated around a thin-shell. Furthermore, the random variables transformed by Lipschitz functions are also concentrated around certain values. Using this phenomenon, it can be shown that rows $\{\mathbf{w}_i\}$ of a random orthogonal matrix $\mathbf{W}$ in high dimension are approximately independent random unit vectors and the inner product $\mathbf{w}_i^T \mathbf{x}$ for thin-shell concentrated vector $\mathbf{x}$ can be shown to be approximately Gaussian. Then from the assumptions that $\phi$ is GPN and $\phi$ and $\phi'$ are Lipschitz continuous, the proofs follow. Each of these steps is rigorously proved in Appendix.

## 3 EXPERIMENTS

We verify our theory on both synthetic and real-world data. More experimental results can be found in Appendix. In short, while very deep neural networks with non-GPN activations show vanishing/exploding gradients, GPN versions show stable gradients and improved trainability in both synthetic and real data. Furthermore, compared to dynamical isometry theory, BSNNs do not exhibit pseudo-linearity and maintain nonlinear functionality.

### 3.1 SYNTHETIC DATA

We create synthetic data to test the norm-preservation properties of the neural networks. The input $\mathbf{x}^1$ is 500 data points of random standard Gaussian vectors of 500 dimension. The gradient error $\partial E / \partial \mathbf{x}^{L+1}$ is also random standard Gaussian vector of 500 dimension. All the neural networks have depth 200. All the weight matrices are random orthogonal matrices uniformly generated. No training is performed.

In Figure 1, we show the norm of inputs and gradient of the neural networks of width 500. From the results, we can see that with GPN, the gradient exploding/vanishing problem is eliminated to large extent. The neural network with Tanh activation function does not show gradient exploding/vanishing problem either. However, $\|\mathbf{x}^{(l)}\|$ is close to zero for large $l$ and each layer is close to a linear one since $\text{Tanh}(x) \approx x$ when $x \approx 0$ (pseudo-linearity), for which dynamical isometry is achieved.

One might wonder if bidirectionally self-noramlizingnormalization has the same effect as dynamical isometry in solving the gradient exploding/vanishing problem, that is, to make the neural network close to an orthogonal matrix. To answer this question, we show the histogram of $\phi'(h_i^{(l)})$ in Figure 2. If the functionality of a neural network is close to an orthogonal matrix, since the weight matrices are orthogonal, then the values of $\phi'(h_i^{(l)})$ would concentrate around one (Figure 2 (a)), which is not the case for BSNNs (Figure 2 (b)). This shows that BSNNs do not suffer from the gradient vanishing/explosion problem while exhibiting nonlinear functionality.

In Figure 7 in Appendix, we show the gradient norm of BSNNs with varying width. There we note that as the width increases, the norm of gradient in each layer of the neural network becomes more equalized, as predicted by our theory.

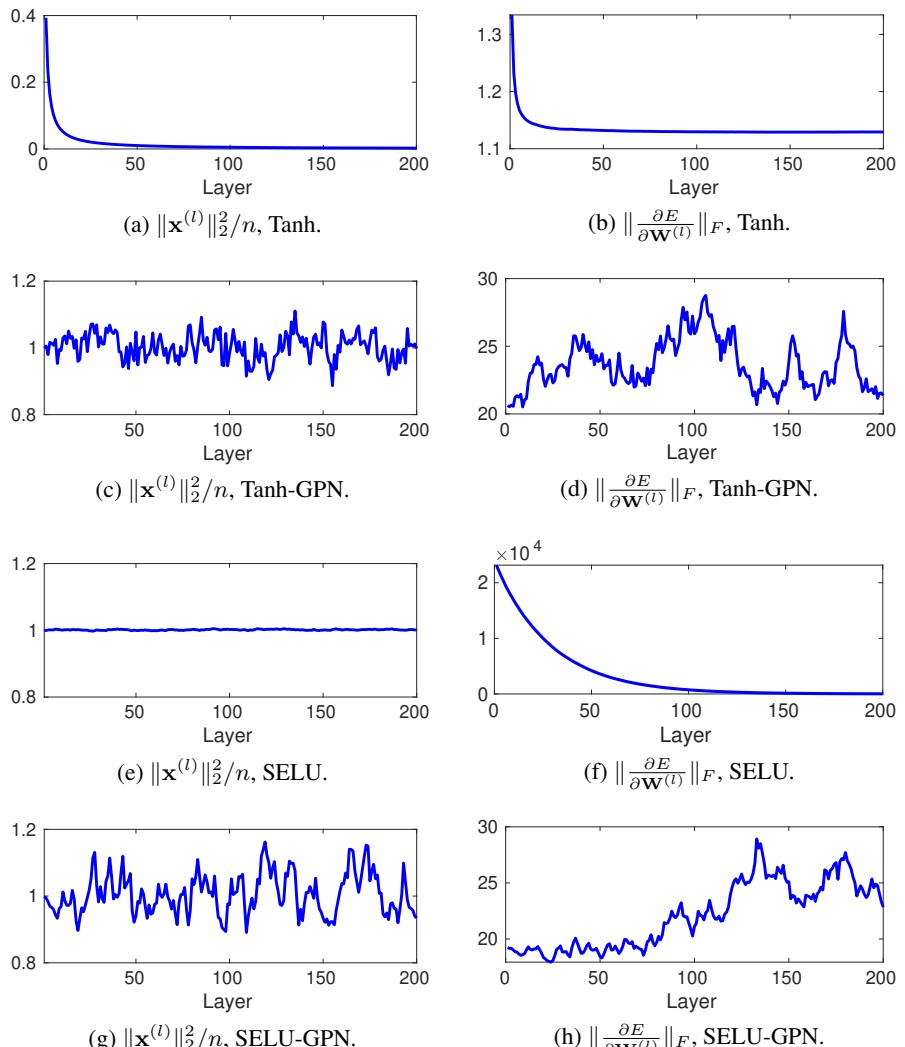

Figure 1: Results on synthetic data with different activation functions. "-GPN" denotes the function is Gaussian-Poincaré normalized. $\|\mathbf{x}^{(l)}\|_2$ denotes the $l_2$ norm of the outputs of the $l$-th layer. $n$ denotes the width. $\|\frac{\partial E}{\partial \mathbf{W}^{(l)}}\|_F$ is the Frobenius norm of the gradient of the weight matrix in the $l$-th layer.

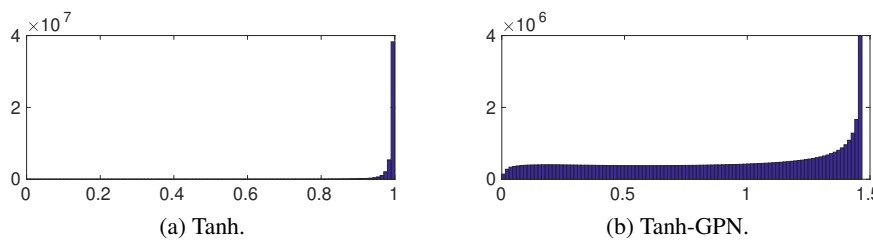

Figure 2: Histogram of $\phi'(h_i^{(l)})$. The values of $\phi'(h_i^{(l)})$ are accumulated for all units, all layers and all samples in the histogram.

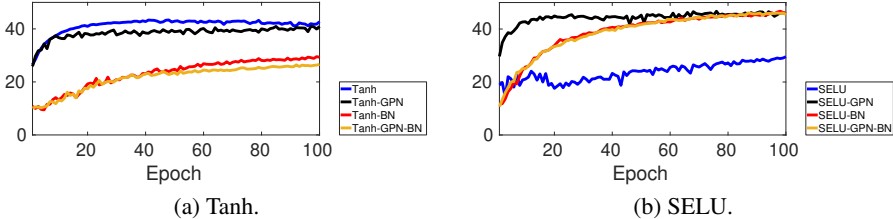

Figure 3: Test accuracy (percentage) during training on CIFAR-10. "-BN" denotes that batch normalization is applied before the activation function.

| | MNIST | | CIFAR-10 | |
|---|---|---|---|---|
| | Train | Test | Train | Test |
| Tanh | 99.05 (87.39) | **96.57** (89.32) | 80.84 (27.90) | **42.71** (29.32) |
| Tanh-GPN | **99.81** (84.93) | 95.54 (87.11) | **96.39** (25.13) | 40.95 (26.58) |
| ReLU | 11.24 (11.24) | 11.35 (11.42) | 10.00 (10.00) | 10.00 (10.00) |
| ReLU-GPN | **33.28** (11.42) | **28.13** (11.34) | **46.60** (10.09) | **34.96** (9.96) |
| LeakyReLU | 11.24 (11.24) | 11.35 (11.63) | 10.00 (10.21) | 10.00 (10.06) |
| LeakyReLU-GPN | **43.17** (11.19) | **49.28** (11.66) | **51.85** (9.89) | **39.38** (10.00) |
| ELU | 99.06 (98.24) | 95.41 (**97.48**) | 80.73 (42.39) | **45.76** (44.16) |
| ELU-GPN | **100.00** (97.86) | 96.56 (96.69) | **99.37** (43.35) | 43.12 (44.36) |
| SELU | 99.86 (97.82) | 97.33 (97.38) | 29.23 (46.47) | 29.55 (45.88) |
| SELU-GPN | **99.92** (97.91) | 96.97 (**97.39**) | **98.24** (47.74) | **45.90** (45.52) |
| GELU | 11.24 (12.70) | 11.35 (10.28) | 10.00 (10.43) | 10.00 (10.00) |
| GELU-GPN | **97.67** (11.22) | **95.82** (9.74) | **90.51** (10.00) | **36.94** (10.00) |

Table 2: Accuracy (percentage) of neural networks of depth 200 with different activation functions on real-world data. The numbers in parenthesis denote the results when batch normalization is applied before the activation function.

## 3.2 REAL-WORLD DATA

We run experiments on real-world image datasets MNIST and CIFAR-10. The neural networks have width 500 and depth 200 (plus one unconstrained layer at bottom and one at top to fit the dimensionality of the input and output). We use stochastic gradient descent of momentum 0.5 with mini-batch size 64 and learning rate 0.0001. The training is run for 50 epochs for MNIST and 100 epochs for CIFAR-10. We do not use data augmentation. Since it is computationally expensive to enforce the orthogonality constraint, we simply constrain each row of the weight matrix to have $l_2$ norm one as a relaxation of orthogonality by the following parametrization $\mathbf{W} = (\mathbf{v}_1/\|\mathbf{v}_1\|_2, \mathbf{v}_2/\|\mathbf{v}_2\|_2, ..., \mathbf{v}_n/\|\mathbf{v}_n\|_2)^T$ and optimize $\mathbf{V} = (\mathbf{v}_1, \mathbf{v}_2, ..., \mathbf{v}_n)^T$ as an unconstrained problem.

We summarize the results in Table 2. We can see that, for activation functions ReLU, LeakyReLU and GELU, the neural networks are not trainable. But once these functions are GPN, the neural network can be trained. GPN activation functions consistently outperform their unnormalized counterparts in terms of the trainability, as the training accuracy is increased, but not necessarily generalization ability. We show the test accuracy during training in Figure 3, from which we can see the training is accelerated when SELU is GPN. ReLU, LeakyReLU and GELU, if not GPN, are completely untrainable due to the vanished gradient (see Appendix).

We observe that batch normalization leads to gradient explosion when combining with any of the activation functions. This confirms the claim of (Philipp et al., 2018) and (Yang et al., 2019) that batch normalization does not solve the gradient exploding/vanishing problem. On the other hand, without batch normalization the neural network with any GPN activation function has stable gradient magnitude throughout training (see Appendix). This indicates that BSNNs can dispense with batch normalization and therefore avoid its shortcomings.

## 4 RELATED WORK

We compare our theory to several most relevant theories in literature. A key distinguishing feature of our theory is that we provide rigorous proofs of the conditions under which the exploding/vanishing problem disappears. To the best of our knowledge, this is the first time that the problem is provably solved for nonlinear neural networks.

### 4.1 SELF-NORMALIZING NEURAL NETWORKS

Self-normalizing neural networks enforce zero mean and unit variance for the output of each unit with the SELU activation function (Klambauer et al., 2017). However, as pointed out in (Philipp et al., 2018) and confirmed in our experiments, only constraining forward signal propagation does not solve the gradient exploding/vanishing problem since the norm of the backward signal can grow or shrink. The signal propagation in both directions needs to be constrained as in our theory.

### 4.2 DEEP SIGNAL PROPAGATION

Our theory is developed from the deep signal propagation theory (Poole et al., 2016; Schoenholz et al., 2017). Both theories require $\mathbb{E}_{x \sim \mathcal{N}(0,1)}[\phi'(x)^2] = 1$. However, ours also requires the quantity $\mathbb{E}_{x \sim \mathcal{N}(0,1)}[\phi(x)^2]$ to be one while in Poole *et al.* (Poole et al., 2016; Schoenholz et al., 2017) it can be an arbitrary positive number. We emphasize that it is desirable to enforce $\mathbb{E}_{x \sim \mathcal{N}(0,1)}[\phi(x)^2] = 1$ to avoid trivial solutions. For example, if $\phi(x) = \text{Tanh}(\epsilon x)$ with $\epsilon \approx 0$, then $\phi(\epsilon x) \approx \epsilon x$ and the neural network becomes essentially a linear one for which depth is unnecessary (pseudo-linearity (Philipp et al., 2018)). This is observed in Figure 1 (a). Moreover, in (Poole et al., 2016; Schoenholz et al., 2017) the signal propagation analysis is done based on random weights under i.i.d. Gaussian distribution whereas we proved how one can solve gradient vanishing/exploding problem assuming the weight matrices are orthogonal and uniformly distributed under Haar measure.

### 4.3 DYNAMICAL ISOMETRY

Dynamical isometry theory (Pennington et al., 2017) enforces the Jacobian matrix of the input-output function of a neural network to have all singular values close to one. Since the weight matrices are constrained to be orthogonal, it is equivalent to enforce each $\mathbf{D}^{(l)}$ in (2) to be close to the identity matrix, which implies the functionality of neural network at initialization is close to an orthogonal matrix (pseudo-linearity). This indeed enables trainability since linear neural networks with orthogonal weight matrices do not suffer from the gradient exploding/vanishing problem. As neural networks need to learn a nonlinear input-output functionality to solve certain tasks, during training the weights of a neural network are unconstrained so that the neural network would move to a nonlinear region where the gradient exploding/vanishing problem might return. In our theory, although the orthogonality of weight matrices is also required, we approach the problem from a different perspective. We do not encourage the linearity at initialization. The neural network can be initialized to be nonlinear and stay nonlinear during the training even when the weights are constrained.

## 5 CONCLUSION

In this paper, we have introduced bidirectionally self-normalizing neural network (BSNN) which constrains both forward and backward signal propagation using a new class of Gaussian-Poincaré normalized activation functions and orthogonal weight matrices. BSNNs are not restrictive in the sense that many commonly used activation functions can be Gaussian-Poincaré normalized. We have rigorously proved that gradient vanishing/exploding problem disappears in BSNNs with high probability under mild conditions. Experiments on synthetic and real-world data confirm the validity of our theory and demonstrate that BSNNs have excellent trainability without batch normalization. Currently, the theoretical analysis is limited to same width, fully-connected neural networks. Future work includes extending our theory to more sophisticated networks such as convolutional architectures as well as investigating the generalization capabilities of BSNNs.

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

## APPENDIX A   PROOFS

**Proposition 1.** *If a neural network is bidirectionally self-normalizing, then*

$$\left\|\frac{\partial E}{\partial \mathbf{W}^{(1)}}\right\|_F = \left\|\frac{\partial E}{\partial \mathbf{W}^{(2)}}\right\|_F = ... = \left\|\frac{\partial E}{\partial \mathbf{W}^{(L)}}\right\|_F. \tag{14}$$

*Proof.* For each $l$, we have

$$\left\|\frac{\partial E}{\partial \mathbf{W}^{(l)}}\right\|_F = \sqrt{\text{trace}\Big(\frac{\partial E}{\partial \mathbf{W}^{(l)}}\Big(\frac{\partial E}{\partial \mathbf{W}^{(l)}}\Big)^T\Big)} \tag{15}$$

$$= \sqrt{\text{trace}(\mathbf{d}^{(l)}(\mathbf{x}^{(l)})^T \mathbf{x}^{(l)}(\mathbf{d}^{(l)})^T)} \tag{16}$$

$$= \sqrt{\text{trace}((\mathbf{x}^{(l)})^T \mathbf{x}^{(l)}(\mathbf{d}^{(l)})^T \mathbf{d}^{(l)})} \tag{17}$$

$$= \sqrt{(\mathbf{x}^{(l)})^T \mathbf{x}^{(l)}}\sqrt{(\mathbf{d}^{(l)})^T \mathbf{d}^{(l)}} \tag{18}$$

$$= \|\mathbf{x}^{(l)}\|_2 \|\mathbf{d}^{(l)}\|_2. \tag{19}$$

By the definition of bidirectional self-normalization, we have $\|\frac{\partial E}{\partial \mathbf{W}^{(1)}}\|_F = ... = \|\frac{\partial E}{\partial \mathbf{W}^{(L)}}\|_F$.  □

**Proposition 2.** *Function $\phi : \mathbb{R} \to \mathbb{R}$ is Gaussian-Poincaré normalized and $\mathbb{E}_{x \sim \mathcal{N}(0,1)}[\phi(x)] = 0$ if and only if $\phi(x) = x$ or $\phi(x) = -x$.*

*Proof.* Since $\mathbb{E}_{x \sim \mathcal{N}(0,1)}[\phi(x)^2] < \infty$ and $\mathbb{E}_{x \sim \mathcal{N}(0,1)}[\phi'(x)^2] < \infty$, $\phi(x)$ and $\phi'(x)$ can be expanded in terms of Hermite polynomials. Let the Hermite polynomial of degree $k$ be

$$H_k(x) = \frac{(-1)^k}{\sqrt{k!}} \exp(\frac{x^2}{2})\frac{d^k}{dx^k}\exp(-\frac{x^2}{2}) \tag{20}$$

and due to $H_k'(x) = \sqrt{k}H_{k-1}(x)$, we have

$$\phi(x) = \sum_{k=0}^{\infty} a_k H_k(x), \tag{21}$$

$$\phi'(x) = \sum_{k=1}^{\infty} \sqrt{k}a_k H_{k-1}(x). \tag{22}$$

Since $\mathbb{E}_{x \sim \mathcal{N}(0,1)}[\phi(x)] = 0$, we have

$$a_0 = \mathbb{E}_{x \sim \mathcal{N}(0,1)}[H_0(x)\phi(x)] \tag{23}$$

$$= \mathbb{E}_{x \sim \mathcal{N}(0,1)}[\phi(x)] \tag{24}$$

$$= 0. \tag{25}$$

Since

$$\mathbb{E}_{x \sim \mathcal{N}(0,1)}[\phi(x)^2] = \mathbb{E}_{x \sim \mathcal{N}(0,1)}[\phi'(x)^2] = 1 \tag{26}$$

and Hermite polynomials are orthonormal, we have

$$\mathbb{E}_{x \sim \mathcal{N}(0,1)}[\phi(x)^2] = \sum_{k=1}^{\infty} a_k^2 = \mathbb{E}_{x \sim \mathcal{N}(0,1)}[\phi'(x)^2] = \sum_{k=1}^{\infty} k a_k^2 = 1. \tag{27}$$

Therefore, we have

$$\sum_{k=1}^{\infty} k a_k^2 - \sum_{k=1}^{\infty} a_k^2 = 0 \tag{28}$$

that is

$$\sum_{k=2}^{\infty} (k-1)a_k^2 = 0. \tag{29}$$

Since each term in $\sum_{k=2}^{\infty}(k-1)a_k^2$ is nonnegative, the only solution is $a_k = 0$ for $k \geq 2$. And since $\mathbb{E}_{x \sim \mathcal{N}(0,1)}[\phi(x)^2] = a_1^2 = 1$, we have $a_1 = \pm 1$. Hence, $\phi(x) = \pm H_1(x) = \pm x$.  □

**Proposition 3.** *For any differentiable function* $\phi : \mathbb{R} \rightarrow \mathbb{R}$ *with non-zero and bounded* $\mathbb{E}_{x \sim \mathcal{N}(0,1)}[\phi(x)^2]$ *and* $\mathbb{E}_{x \sim \mathcal{N}(0,1)}[\phi'(x)^2]$, *there exist two constants* $a$ *and* $b$ *such that* $a\phi(x) + b$ *is Gaussian-Poincaré normalized.*

*Proof.* Let $\varphi(x) = \phi(x) + c$. Then let

$$\psi(c) = \mathbb{E}_{x \sim \mathcal{N}(0,1)}[\varphi(x)^2] - \mathbb{E}_{x \sim \mathcal{N}(0,1)}[(\phi'(x))^2] \tag{30}$$

$$= \text{Var}_{x \sim \mathcal{N}(0,1)}[\varphi(x)] + (\mathbb{E}_{x \sim \mathcal{N}(0,1)}[\varphi(x)])^2 - \mathbb{E}_{x \sim \mathcal{N}(0,1)}[(\phi'(x))^2] \tag{31}$$

$$= \text{Var}_{x \sim \mathcal{N}(0,1)}[\phi(x)] + (\mathbb{E}_{x \sim \mathcal{N}(0,1)}[\phi(x)] + c)^2 - \mathbb{E}_{x \sim \mathcal{N}(0,1)}[(\phi'(x))^2]. \tag{32}$$

Therefore, $\psi(c)$ is a quadratic function of $c$. We also have $\psi(c) > 0$ as $c \rightarrow \infty$ and $\psi(-\mathbb{E}_{x \sim \mathcal{N}(0,1)}[\phi(x)]) \leq 0$ due to Gaussian-Poincaré inequality. Hence, there exists $c$ for which $\psi(c) = 0$ such that $\mathbb{E}_{x \sim \mathcal{N}(0,1)}[(\phi(x)+c)^2] = \mathbb{E}_{x \sim \mathcal{N}(0,1)}[\phi'(x)^2]$. Let $a = (\mathbb{E}_{x \sim \mathcal{N}(0,1)}[\phi'(x)^2])^{-1/2}$ and $b = ac$, we have $\mathbb{E}_{x \sim \mathcal{N}(0,1)}[(a\phi(x) + b)^2] = \mathbb{E}_{x \sim \mathcal{N}(0,1)}[(a\phi'(x))^2] = 1$. $\square$

The proof is largely due to (Eldredge, 2020) with minor modification in here.

**Assumptions.**

1. *Random vector* $\mathbf{x} \in \mathbb{R}^n$ *is thin-shell concentrated.*
2. *Random orthogonal matrix* $\mathbf{W} = (\mathbf{w}_1, \mathbf{w}_2, ..., \mathbf{w}_n)^T$ *is uniformly distributed.*
3. *Function* $\phi : \mathbb{R} \rightarrow \mathbb{R}$ *is Gaussian-Poincaré normalized.*
4. *Function* $\phi$ *and its derivative are Lipschitz continuous.*

**Theorem 2** (**Forward Norm-Preservation**). *Random vector*

$$(\phi(\mathbf{w}_1^T \mathbf{x}), \phi(\mathbf{w}_2^T \mathbf{x}), ..., \phi(\mathbf{w}_n^T \mathbf{x})) \tag{33}$$

*is thin-shell concentrated.*

**Theorem 3** (**Backward Norm-Preservation**). *Let* $\mathbf{D}$ *be the diagonal matrix whose diagonal elements are* $D_{ii} = \phi'(\mathbf{w}_i^T \mathbf{x})$ *and* $\mathbf{y} \in \mathbb{R}^n$ *be a fixed vector with bounded* $\|\mathbf{y}\|_\infty$. *Then for any* $\epsilon > 0$

$$\mathbb{P}\Big\{\frac{1}{n}\Big|\|\mathbf{D}\mathbf{y}\|_2^2 - \|\mathbf{y}\|_2^2\Big| \geq \epsilon\Big\} \rightarrow 0 \tag{34}$$

*as* $n \rightarrow \infty$.

**Notations.** $\mathbb{S}^{n-1} = \{\mathbf{x} \in \mathbb{R}^n : \|\mathbf{x}\|_2 = 1\}$. $\mathbb{O}(n)$ is the orthogonal matrix group of size $n$. $\mathbf{1}_{\{\cdot\}}$ denotes the indicator function. $\mathbf{0}_n$ denotes the vector of dimension $n$ and all elements equal to zero. $\mathbf{I}_n$ denotes the identity matrix of size $n \times n$.

**Lemma 1.** *If random variable* $x \sim \mathcal{N}(0, 1)$ *and function* $f : \mathbb{R} \rightarrow \mathbb{R}$ *is Lipschitz continuous, then random variable* $f(x)$ *is sub-gaussian.*

*Proof.* Due to the Gaussian concentration theorem (Theorem 5.2.2 in (Vershynin, 2018)), we have

$$\|f(x) - \mathbb{E}[f(x)]\|_{\psi_2} \leq CK \tag{35}$$

where $\|\cdot\|_{\psi_2}$ denotes sub-gaussian norm, $C$ is a constant and $K$ is the Lipschitz constant of $f$. This implies $f(x) - \mathbb{E}[f(x)]$ is sub-gaussian (Proposition 2.5.2 in (Vershynin, 2018)). Therefore $f(x)$ is sub-gaussian (Lemma 2.6.8 in (Vershynin, 2018)). $\square$

**Lemma 2.** *Let* $\mathbf{x} = (x_1, x_2, ..., x_n) \in \mathbb{R}^n$ *be a random vector that each coordinate* $x_i$ *is independent and sub-gaussian and* $\mathbb{E}[x_i^2] = 1$. *Let* $\mathbf{y} = (y_1, y_2, ..., y_n) \in \mathbb{R}^n$ *be a fixed vector with bounded* $\|\mathbf{y}\|_\infty$. *Then*

$$\mathbb{P}\Big\{\frac{1}{n}\Big|\sum_i x_i^2 y_i^2 - \sum_i y_i^2\Big| \geq \epsilon\Big\} \rightarrow 0 \tag{36}$$

*as* $n \rightarrow \infty$.

*Proof.* Since $y_i x_i$ is sub-gaussian, then $y_i^2 x_i^2$ is sub-exponential (Lemma 2.7.6 in (Vershynin, 2018)). Since $\mathbb{E}[y_i^2 x_i^2] = y_i^2 \mathbb{E}[x_i^2] = y_i^2$, $y_i^2 x_i^2 - y_i^2$ is sub-exponential with zero mean (Exercise 2.7.10 in (Vershynin, 2018)). Applying Bernsteins inequality (Corollary 2.8.3 in (Vershynin, 2018)), we proved the lemma. □

**Lemma 3.** *Let $\mathbf{z} \sim \mathcal{N}(\mathbf{0}_n, \mathbf{I}_n)$. Then for any $0 < \delta < 1$*

$$\mathbb{P}\{\mathbf{z} \in \mathbb{R}^n : (1-\delta)\sqrt{n} \leq \|\mathbf{z}\|_2 \leq (1+\delta)\sqrt{n}\} \geq 1 - 2\exp(-n\delta^2). \tag{37}$$

See (Alberts & Khoshnevisan, 2018) (Theorem 1.2) for a proof.

**Lemma 4.** *Let $\mathbf{z} \sim \mathcal{N}(\mathbf{0}_n, \mathbf{I}_n)$. Then $\mathbf{z}/\|\mathbf{z}\|_2$ is uniformly distributed on $\mathbb{S}^{n-1}$.*

See (Dawkins, 2016) for a proof.

**Lemma 5.** *Let $\mathbf{z} = (z_1, z_2, ..., z_n) \sim \mathcal{N}(\mathbf{0}_n, \mathbf{I}_n)$, $\mathbf{a} = (a_1, a_2, ..., a_n)$ be a fixed vector with bounded $\|\mathbf{a}\|_\infty$ and $f : \mathbb{R} \to \mathbb{R}$ be a continuous function. Then for any $\epsilon > 0$*

$$\mathbb{P}\left\{\frac{1}{n}\left|\sum_i y_i f(\sqrt{n}/\|\mathbf{z}\|_2 z_i) - \sum_i y_i f(z_i)\right| > \epsilon\right\} \to 0 \tag{38}$$

*as $n \to \infty$.*

*Proof.* Since

$$\frac{1}{n}\left|\sum_i a_i f(\sqrt{n}/\|\mathbf{z}\|_2 z_i) - \sum_i a_i f(z_i)\right| \leq \frac{1}{n}\sum_i |a_i| \cdot |f(\sqrt{n}/\|\mathbf{z}\|_2 z_i) - f(z_i)|, \tag{39}$$

if, as $n \to \infty$,

$$\mathbb{P}\left\{\frac{1}{n}\sum_i |a_i| \cdot |f(\sqrt{n}/\|\mathbf{z}\|_2 z_i) - f(z_i)| > \epsilon\right\} \to 0, \tag{40}$$

then

$$\mathbb{P}\left\{\frac{1}{n}\left|\sum_i y_i f(\sqrt{n}/\|\mathbf{z}\|_2 z_i) - \sum_i y_i f(z_i)\right| > \epsilon\right\} \to 0. \tag{41}$$

For $0 < \delta < 1$, let

$$A = \left\{\mathbf{z} \in \mathbb{R}^n : \frac{1}{n}\sum_i |a_i| \cdot |f(\sqrt{n}/\|\mathbf{z}\|_2 z_i) - f(z_i)| > \epsilon\right\}, \tag{42}$$

$$\mathcal{U}_\delta = \left\{\mathbf{z} \in \mathbb{R}^n : (1-\delta)\sqrt{n} \leq \|\mathbf{z}\|_2 \leq (1+\delta)\sqrt{n}\right\}. \tag{43}$$

Then

$$\mathbb{P}\left\{\frac{1}{n}\sum_i |a_i| \cdot |f(\sqrt{n}/\|\mathbf{z}\|_2 z_i) - f(z_i)| > \epsilon\right\} = \int_{\mathbb{R}^n} \mathbf{1}_{\{\mathbf{z} \in A\}} d\mathbf{z} \tag{44}$$

$$= \int_{\mathbb{R}^n \setminus \mathcal{U}_\delta} \mathbf{1}_{\{\mathbf{z} \in A\}} d\mathbf{z} + \int_{\mathcal{U}_\delta} \mathbf{1}_{\{\mathbf{z} \in A\}} d\mathbf{z}. \tag{45}$$

Let $\delta = n^{-1/4}$. From Lemma 3, we have, as $n \to \infty$,

$$\int_{\mathbb{R}^n \setminus \mathcal{U}_\delta} \mathbf{1}_{\{\mathbf{z} \in A\}} d\mathbf{z} \leq \int_{\mathbb{R}^n \setminus \mathcal{U}_\delta} d\mathbf{z} = 1 - \mathbb{P}\{\mathbf{z} \in \mathcal{U}_\delta\} \leq 2\exp(-n\delta^2) \to 0. \tag{46}$$

For $\mathbf{z} \in \mathcal{U}_\delta$ and $\delta = n^{-1/4}$, we have $\|\mathbf{z}\|_2 \to \sqrt{n}$, $\sqrt{n}/\|\mathbf{z}\|_2 z_i \to z_i$ and therefore $f(\sqrt{n}/\|\mathbf{z}\|_2 z_i) \to f(z_i)$ as $n \to \infty$. Hence, $\int_{\mathcal{U}_\delta} \mathbf{1}_{\{\mathbf{z} \in A\}} d\mathbf{z} \to 0$, as $n \to \infty$.

□

**Lemma 6.** *Let random matrix $\mathbf{W}$ be uniformly distributed on $\mathbb{O}(n)$ random vector $\boldsymbol{\theta}$ be uniformly distributed on $\mathbb{S}^{n-1}$ and random vector $\mathbf{x} \in \mathbb{R}^n$ be thin-shell concentrated. Then $\mathbf{W}\mathbf{x} \to \sqrt{n}\boldsymbol{\theta}$ as $n \to \infty$.*

*Proof.* Let $\mathbf{y} \in \mathbb{R}^n$ be any vector with $\|\mathbf{y}\|_2 = \sqrt{n}$ and $\mathbf{e} = (\sqrt{n}, 0, ..., 0) \in \mathbb{R}^n$. Since $\mathbf{W}$ is uniformly distributed, $\mathbf{W}\mathbf{y}$ has the same distribution as $\mathbf{W}\mathbf{e}$. $\mathbf{W}\mathbf{e}$ is the first row of $\sqrt{n}\mathbf{W}$, which is equivalent to random vector $\sqrt{n}\boldsymbol{\theta}$. Since $\mathbf{x}$ is thin-shell concentrated, $\mathbf{x} \to \frac{\sqrt{n}}{\|\mathbf{x}\|_2}\mathbf{x} = \mathbf{y}$ and therefore $\mathbf{W}\mathbf{x} \to \sqrt{n}\boldsymbol{\theta}$ as $n \to \infty$.

$\square$

*Proof of Theorem 2.* Let $\mathbf{z} = (z_1, z_2, ..., z_n) \sim \mathcal{N}(\mathbf{0}, \mathbf{I})$. Due to Lemma 1, random variable $\phi(z_i)$ is sub-gaussian. Since $\phi$ is Gaussian-Poincaré normalized, $\mathbb{E}_{z_i \sim \mathcal{N}(0,1)}[\phi(z_i)^2] = 1$. Applying Lemma 2 with each $y_i = 1$, we have for $\epsilon > 0$

$$\mathbb{P}\left\{\left|\frac{1}{n}\sum_i \phi(z_i)^2 - 1\right| \geq \epsilon\right\} \to 0 \tag{47}$$

as $n \to \infty$.

Due to Lemma 4 and 5 (with each $a_i = 1$), for random vector $\boldsymbol{\theta} = (\theta_1, \theta_2, ..., \theta_n)$ uniformly distributed on $\mathbb{S}^{n-1}$, we have

$$\mathbb{P}\left\{\left|\frac{1}{n}\sum_i \phi(\sqrt{n}\theta_i)^2 - \frac{1}{n}\sum_i \phi(z_i)^2\right| \geq \epsilon\right\} \to 0 \tag{48}$$

and therefore

$$\mathbb{P}\left\{\left|\frac{1}{n}\sum_i \phi(\sqrt{n}\theta_i)^2 - 1\right| \geq \epsilon\right\} \to 0 \tag{49}$$

as $n \to \infty$.

Then from Lemma 6, we have $\mathbf{W}\mathbf{x} \to \sqrt{n}\boldsymbol{\theta}$ and therefore

$$\mathbb{P}\left\{\left|\frac{1}{n}\sum_i \phi(\mathbf{w}_i^T\mathbf{x})^2 - 1\right| \geq \epsilon\right\} \to 0 \tag{50}$$

as $n \to \infty$. $\square$

*Proof of Theorem 3.* Let $\mathbf{z} = (z_1, z_2, ..., z_n)$ be the standard multivariate Gaussian random vectors. Due to Lemma 1, random variable $\phi'(z_i)$ is sub-gaussian. Since $\phi$ is Gaussian-Poincaré normalized, $\mathbb{E}_{z_i \sim \mathcal{N}(0,1)}[\phi'(z_i)^2] = 1$. Applying Lemma 2, we have

$$\mathbb{P}\left\{\frac{1}{n}\left|\sum_i y_i^2 \phi'(z_i)^2 - y_i^2\right| \geq \epsilon\right\} \to 0 \tag{51}$$

as $n \to \infty$.

Due to Lemma 4 and 5 (with each $a_i = y_i^2$), for random vector $\boldsymbol{\theta} = (\theta_1, \theta_2, ..., \theta_n)$ uniformly distributed on $\mathbb{S}^{n-1}$, we have

$$\mathbb{P}\left\{\left|\frac{1}{n}\sum_i y_i^2 \phi'(\sqrt{n}\theta_i)^2 - \frac{1}{n}\sum_i y_i^2 \phi'(z_i)^2\right| \geq \epsilon\right\} \to 0 \tag{52}$$

and therefore

$$\mathbb{P}\left\{\left|\frac{1}{n}\sum_i y_i^2 \phi'(\sqrt{n}\theta_i)^2 - y_i^2\right| \geq \epsilon\right\} \to 0 \tag{53}$$

as $n \to \infty$.

Then from Lemma 6, we have $\mathbf{W}\mathbf{x} \to \sqrt{n}\boldsymbol{\theta}$ and therefore

$$\mathbb{P}\left\{\left|\frac{1}{n}\sum_i y_i^2 \phi'(\mathbf{w}_i^T\mathbf{x})^2 - y_i^2\right| \geq \epsilon\right\} \to 0 \tag{54}$$

as $n \to \infty$.

$\square$

# APPENDIX B  ADDITIONAL EXPERIMENTS

Due to the space limitation, we only showed the experiments with Tanh and SELU activation functions in the main text. In this section, we show the experiments with ReLU, LeakyReLU, ELU and SELU. Additionally, we also measure the gradient exploding/vanishing during training on the real-world data.

## B.1  SYNTHETIC DATA

We show the figures of the experimental results in addition to the ones in the main text.

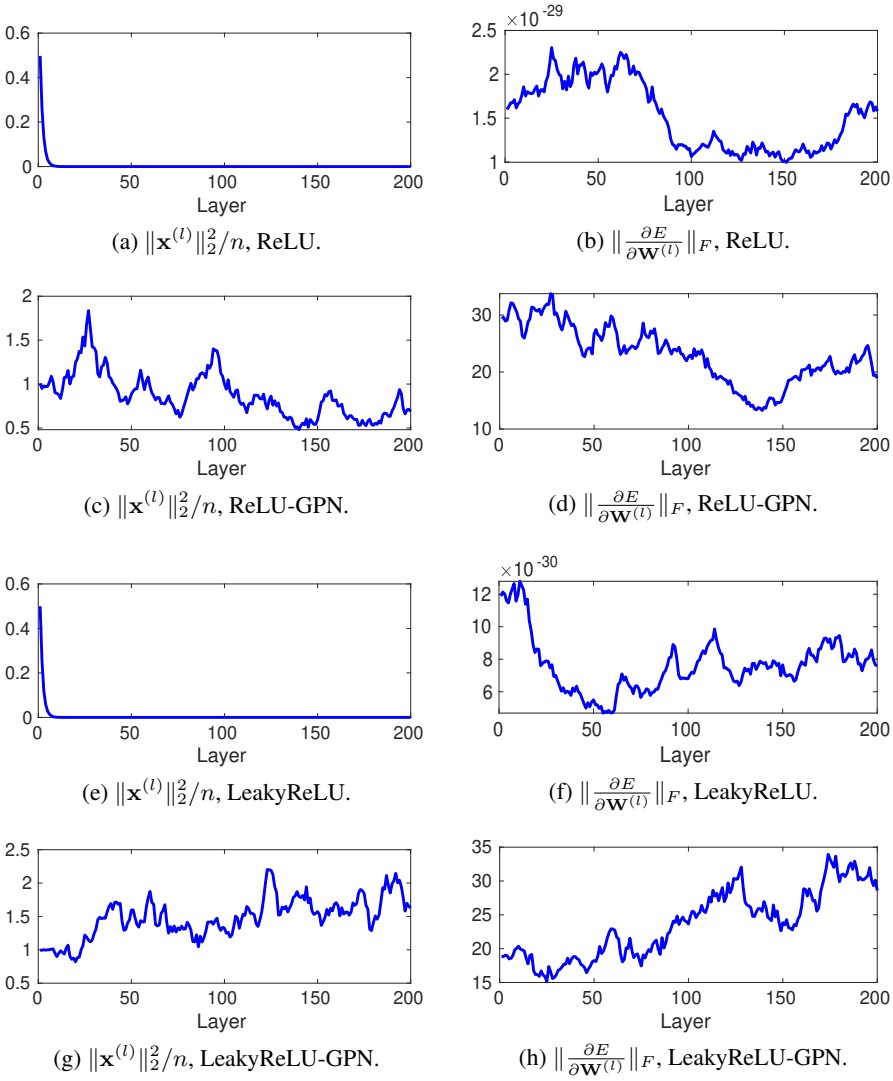

Figure 4: Results on synthetic data with different activation functions. "-GPN" denotes the function is Gaussian-Poincaré normalized. $\|\mathbf{x}^{(l)}\|_2$ denotes the $l_2$ norm of the outputs of the $l$-th layer. $n$ denotes the width. $\|\frac{\partial E}{\partial \mathbf{W}^{(l)}}\|_F$ is the Frobenius norm of the gradient of the weight matrix in the $l$-th layer.

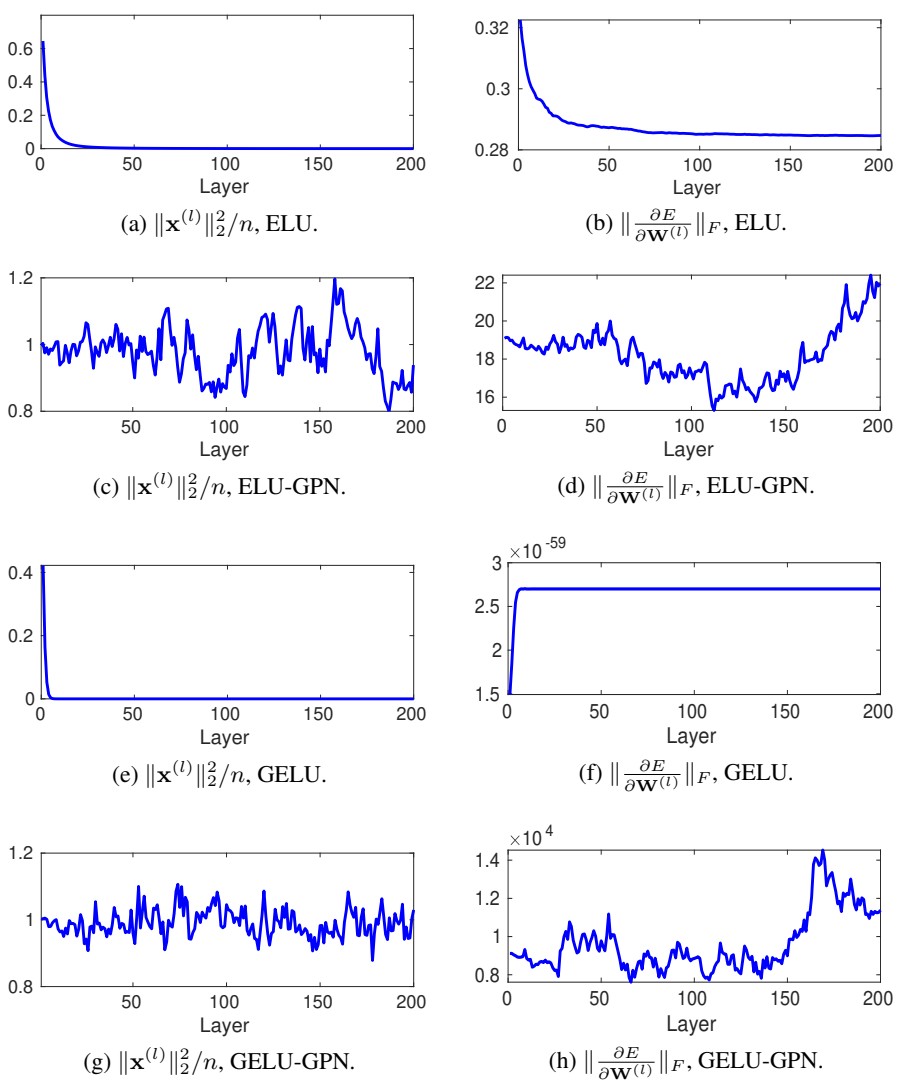

Figure 5: Results on synthetic data with different activation functions. "-GPN" denotes the function is Gaussian-Poincaré normalized. $\|\mathbf{x}^{(l)}\|_2$ denotes the $l_2$ norm of the outputs of the $l$-th layer. $n$ denotes the width. $\|\frac{\partial E}{\partial \mathbf{W}^{(l)}}\|_F$ is the Frobenius norm of the gradient of the weight matrix in the $l$-th layer.

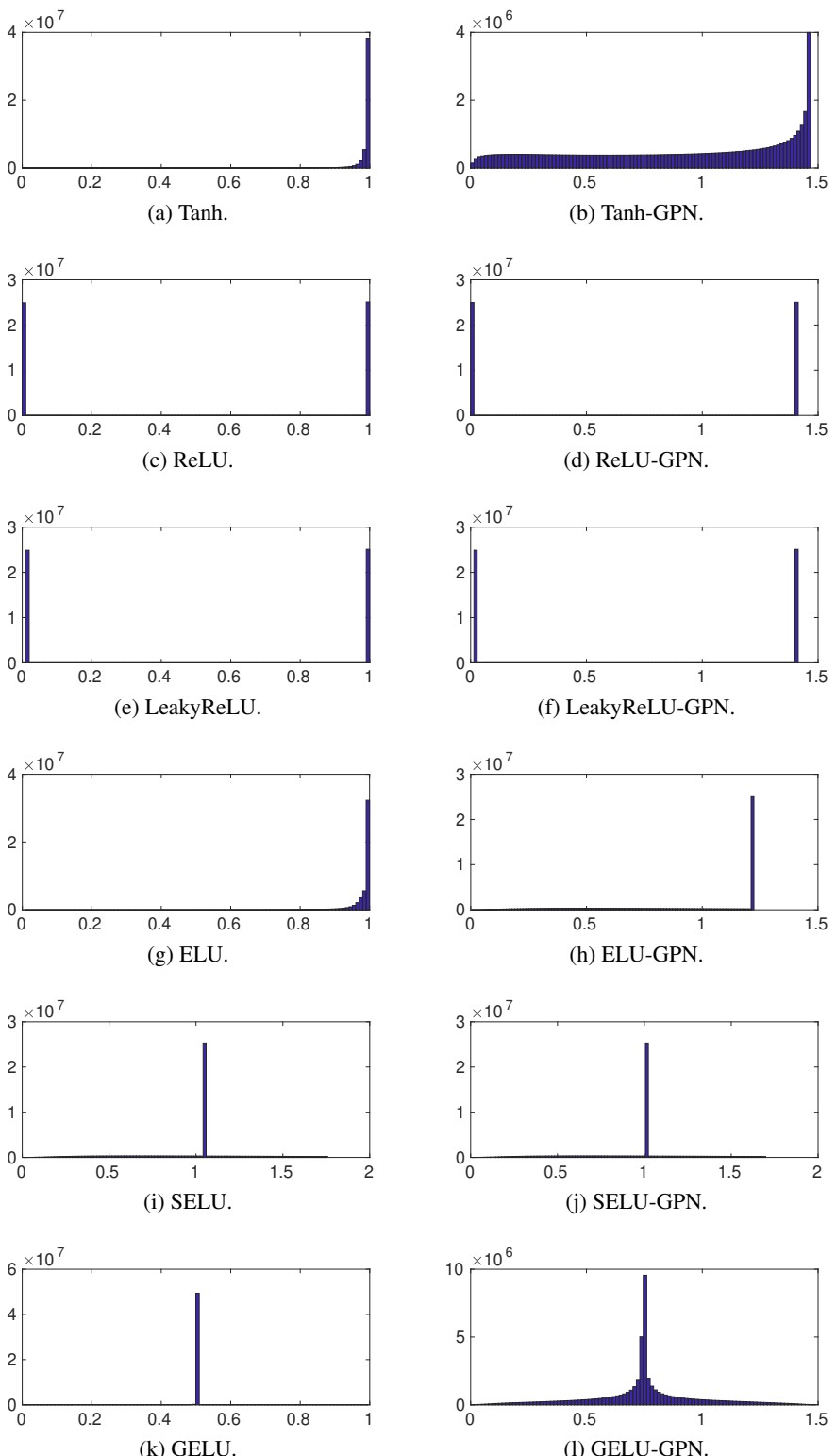

Figure 6: Histogram of $\phi'(h_i^{(l)})$. The values of $\phi'(h_i^{(l)})$ are accumulated for all units, all layers and all samples in the histogram. Except for ELU, none of them has values concentrated around one.

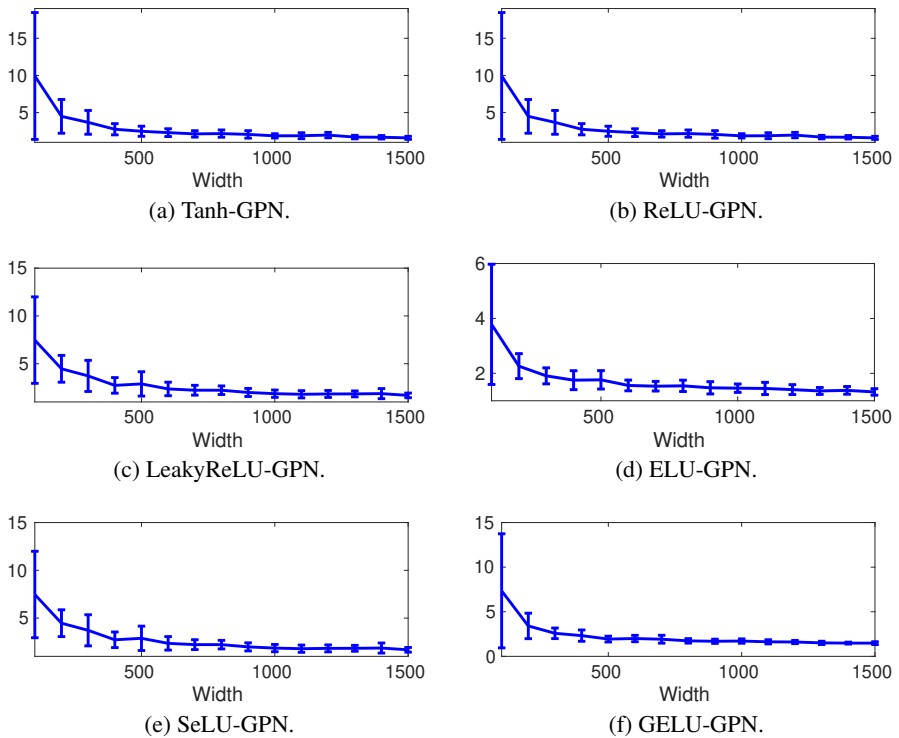

Figure 7: Gradient norm ratio for different layer width on synthetic data. The ratio is defined as $\max_l \|\frac{\partial E}{\partial \mathbf{W}^{(l)}}\|_F / \min_l \|\frac{\partial E}{\partial \mathbf{W}^{(l)}}\|_F$. The width ranges from 100 to 1500. The error bars show standard deviation.

## B.2 REAL-WORLD DATA

In Figure 8, we show the test accuracy during training on MNIST. In Figure 9, we show the test accuracy during training on CIFAR-10.

In Figure 10, 11, 12 and 13, we show a measure of gradient exploding/vanishing during training for different activation functions. The measure is defined as the ratio of the maximum gradient norm and the minimum gradient norm across layers. Since we use the parametrization

$$\mathbf{W} = \left(\frac{\mathbf{v}_1}{\|\mathbf{v}_1\|_2}, \frac{\mathbf{v}_2}{\|\mathbf{v}_2\|_2}, ..., \frac{\mathbf{v}_n}{\|\mathbf{v}_n\|_2}\right)^T \tag{55}$$

with $\mathbf{V} = (\mathbf{v}_1, \mathbf{v}_2, ..., \mathbf{v}_n)^T$, the gradient norm ratio is defined on the unconstrained weights $\mathbf{V}$, that is,

$$\frac{\max_l \|\frac{\partial E}{\partial \mathbf{V}^{(l)}}\|_F}{\min_l \|\frac{\partial E}{\partial \mathbf{V}^{(l)}}\|_F}. \tag{56}$$

Note that for ReLU, LeakyReLU and GELU, the gradient vanishes during training in some experiments and therefore the plots are empty. From the figures, we can see that batch normalization leads to gradient explosion especially at the early stage of training. On the other hand, without batch normalization, the gradient is stable throughout training for GPN activation functions.

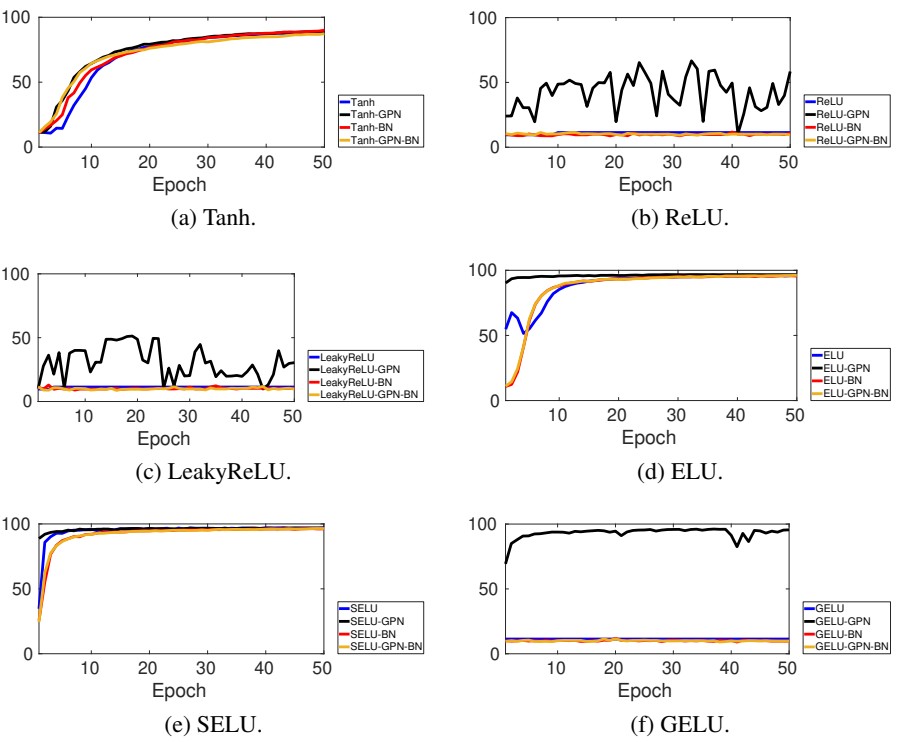

Figure 8: Test accuracy (percentage) during training on MNIST.

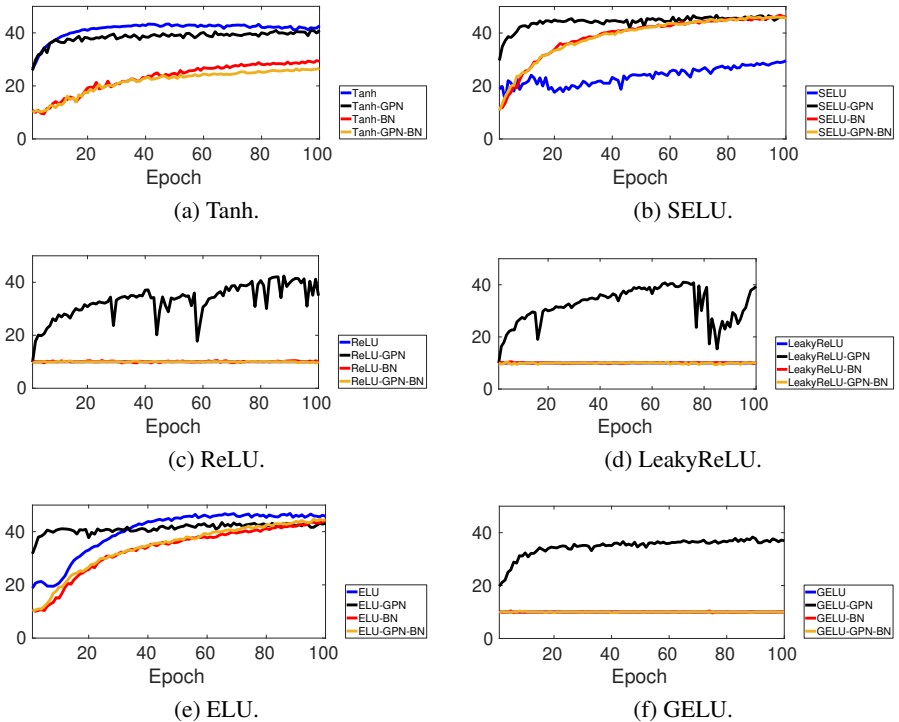

Figure 9: Test accuracy (percentage) during training on CIFAR-10.

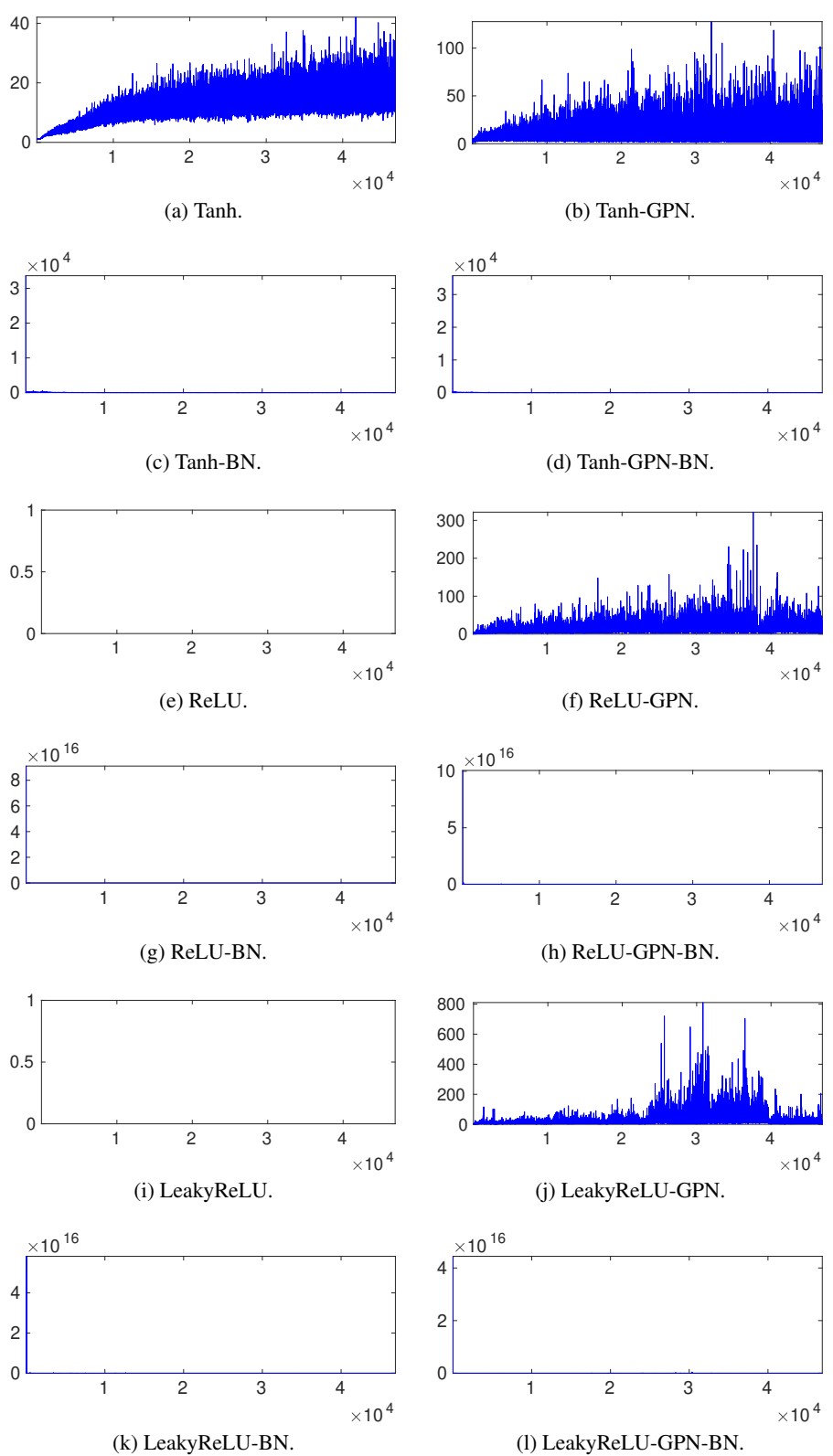

Figure 10: Gradient norm ratio during training on MNIST. Horizontal axis denotes the mini-batch updates. Vertical axis denotes the gradient norm ratio $\max_l \| \frac{\partial E}{\partial \mathbf{V}^{(l)}} \|_F / \min_l \| \frac{\partial E}{\partial \mathbf{V}^{(l)}} \|_F$. The gradient vanishes ($\| \frac{\partial E}{\partial \mathbf{V}^{(l)}} \|_F \approx 0$) for ReLU and LeakyReLU during training and hence the plots are empty.

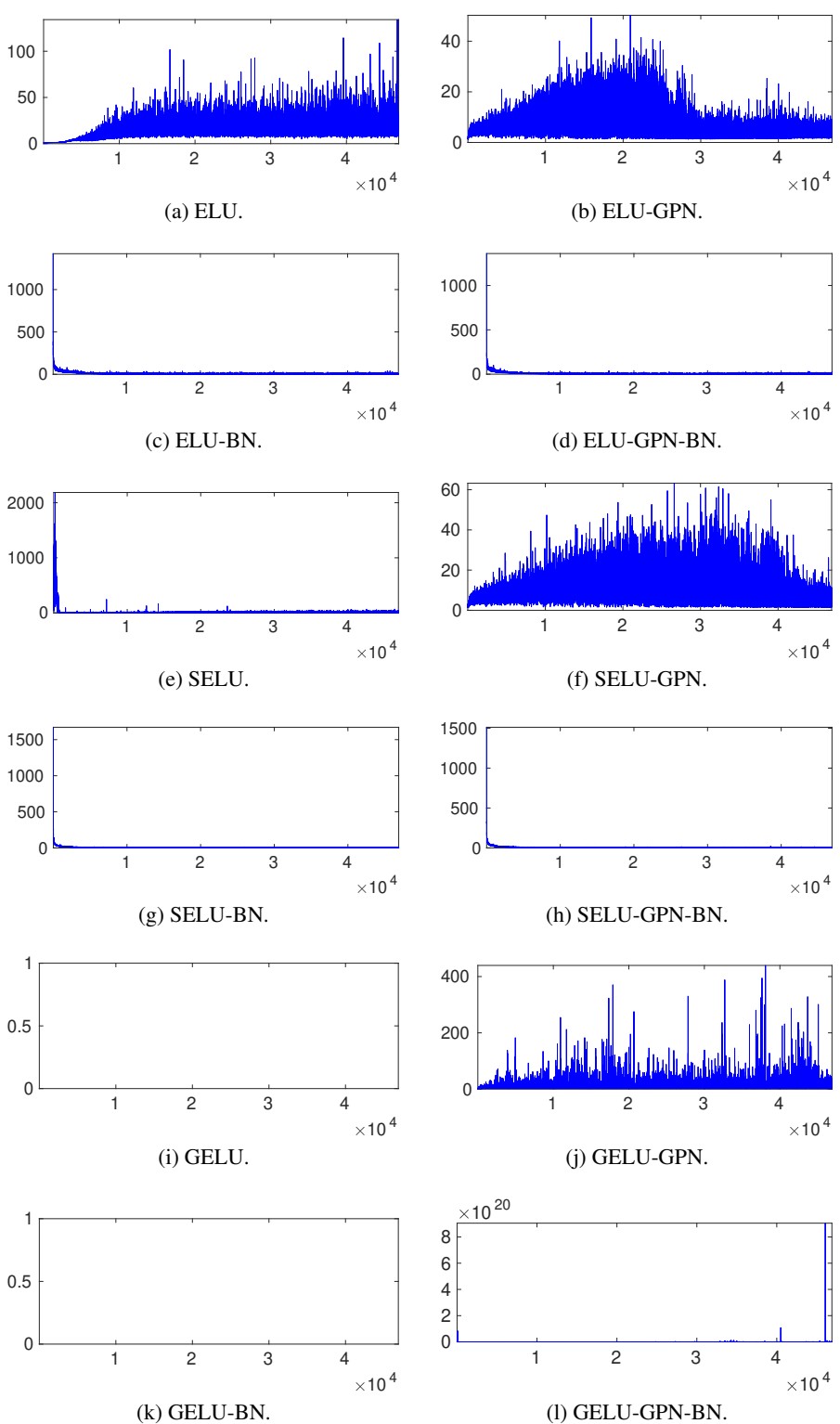

Figure 11: Gradient norm ratio during training on MNIST. Horizontal axis denotes the mini-batch updates. Vertical axis denotes the gradient norm ratio $\max_l \|\frac{\partial E}{\partial \mathbf{V}^{(l)}}\|_F / \min_l \|\frac{\partial E}{\partial \mathbf{V}^{(l)}}\|_F$. The gradient vanishes ($\|\frac{\partial E}{\partial \mathbf{V}^{(l)}}\|_F \approx 0$) for GELU and GELU-BN during training and hence the plots are empty.

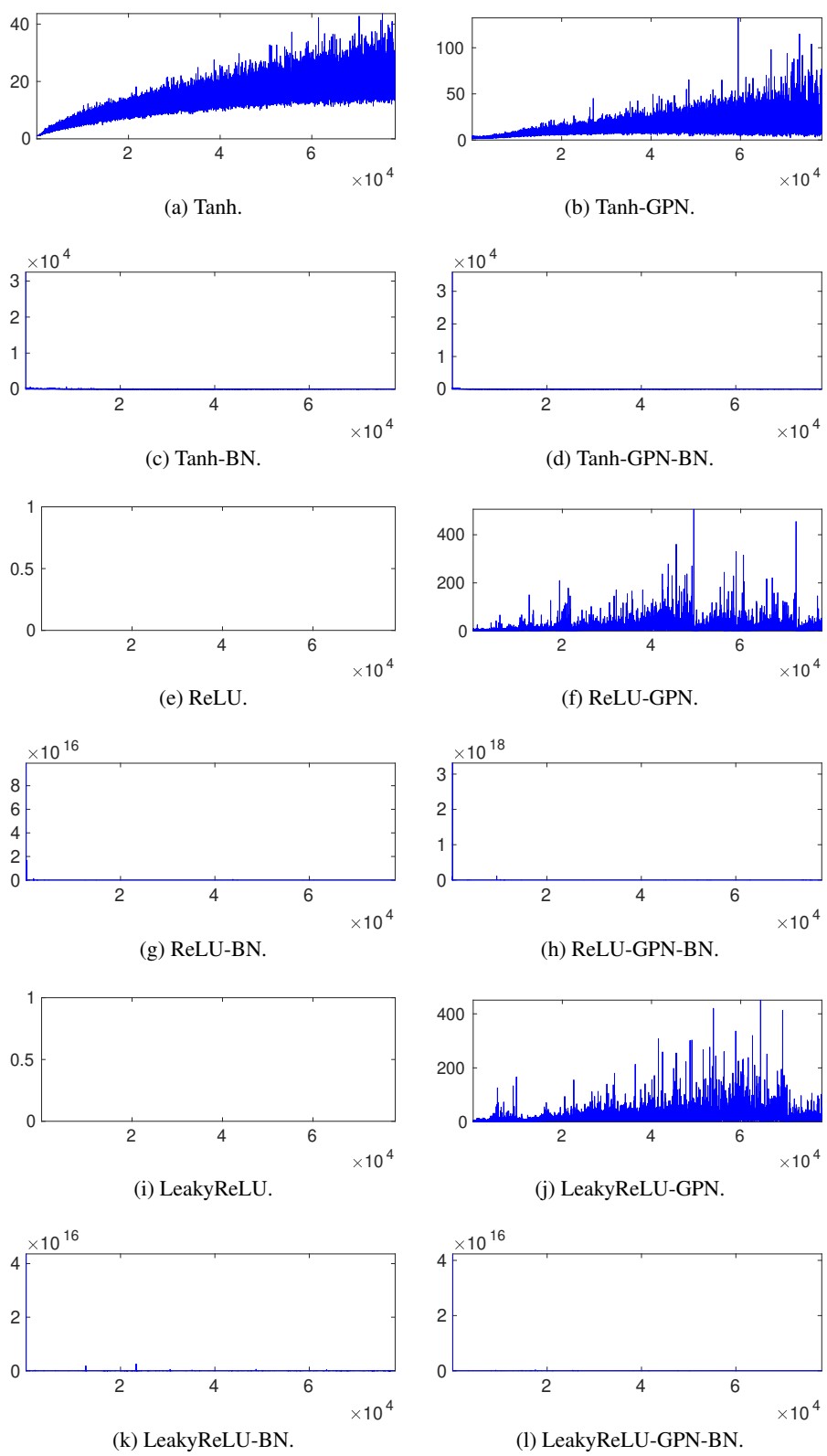

Figure 12: Gradient norm ratio during training on CIFAR-10. Horizontal axis denotes the mini-batch updates. Vertical axis denotes the gradient norm ratio $\max_l \|\frac{\partial E}{\partial \mathbf{V}^{(l)}}\|_F / \min_l \|\frac{\partial E}{\partial \mathbf{V}^{(l)}}\|_F$. The gradient vanishes ($\|\frac{\partial E}{\partial \mathbf{V}^{(l)}}\|_F \approx 0$) for ReLU and LeakyReLU during training and hence the plots are empty.

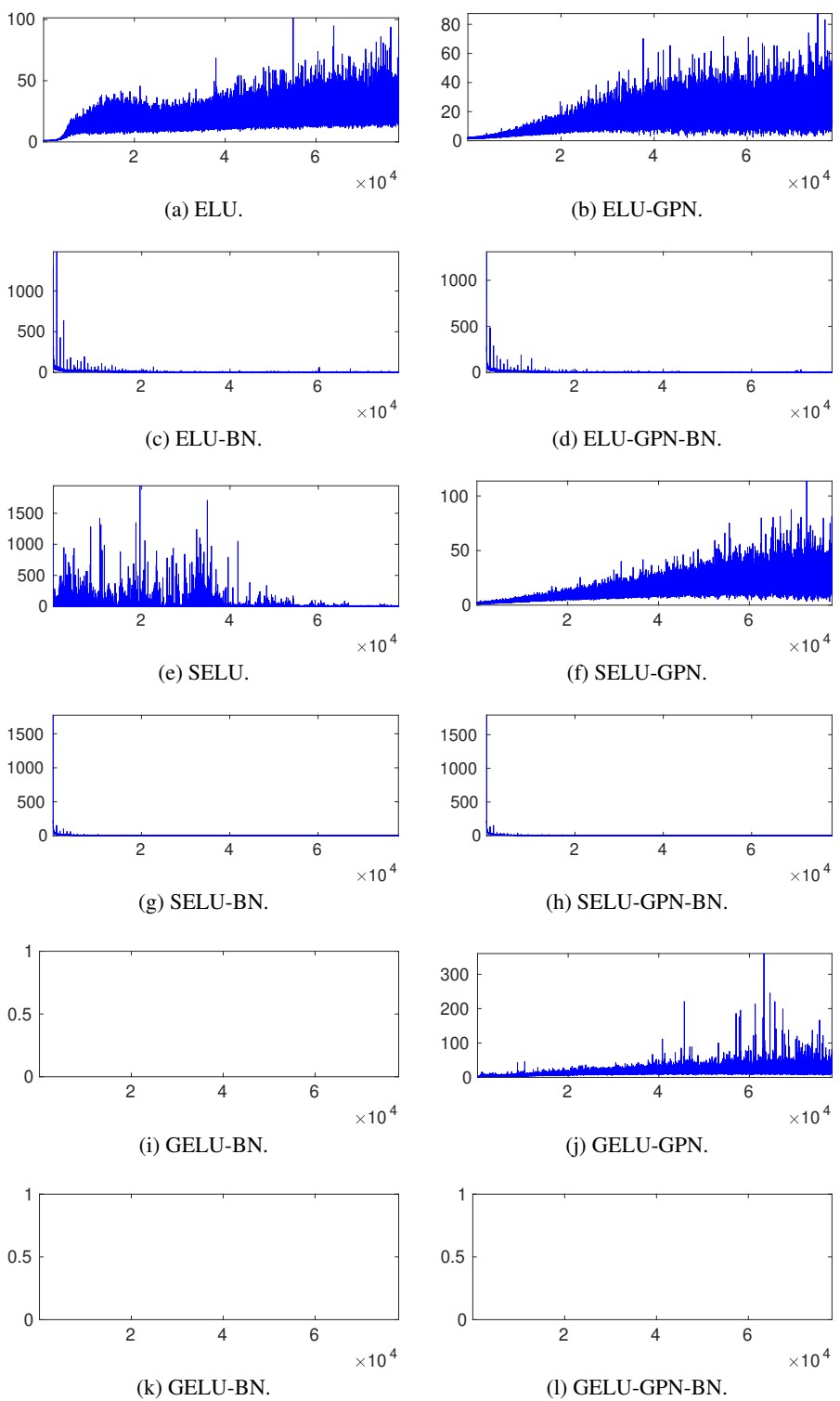

Figure 13: Gradient norm ratio during training on CIFAR-10. Horizontal axis denotes the mini-batch updates. Vertical axis denotes the gradient norm ratio $\max_l \|\frac{\partial E}{\partial \mathbf{V}^{(l)}}\|_F / \min_l \|\frac{\partial E}{\partial \mathbf{V}^{(l)}}\|_F$. The gradient vanishes ($\|\frac{\partial E}{\partial \mathbf{V}^{(l)}}\|_F \approx 0$) for GELU, GELU-BN and GELU-GPN-BN during training and hence the plots are empty. For GELU-GPN-BN, both gradient exploding and gradient vanishing are observed.

