# OpenReview forum: "Bidirectionally Self-Normalizing Neural Networks"
_ICLR.cc/2021/Conference — Reject_

### Official Review · AnonReviewer4 · 2020-10-27
**Nice theoretical treatment, but results equivalent to linearising deep models**

**Rating:** 4
**Confidence:** 4

**Review:**

The paper tackles the problem of vanishing/exploding gradient in neural networks by imposing constraints on the weights and activation functions that ensure consistent Frobenius norm across updates to the weight matrices in all layers of the network.

The paper is very well written, straight to follow, theory is well laid out and it is interesting.  The only (minor) issue I have on the presentation side is the name “bidirectional self-normalizing neural networks”.  This phrasing initially made me think the the network is bidirectional…but the network itself is not, it’s the self-normalizing that is bidirectional.   Wouldn’t it be more precise to say “bidirectionally self-normalized neural networks”?

However, despite a well outlined theory, I think that in the end the experimental evaluation demonstrates a major weakness of the proposed approach.  Credit to the authors for including Table 2 with evaluation on real datasets, but the generalisation performance there seems to show quite clearly that -GPN variants of the networks are equivalent to linear classifiers.  A quick check of the performance of linear classifier on MNIST and CIFAR10 without augmentation reveals 93% and 41% testing accuracy - pretty close to the performance shown in Table 2.  Sure, -GPNising allows for training a 200-layer model, but it seem to come at the cost of reducing the model to barely more than a linear classifier.  The sacrifice of all non-trivial representational power in order to gain stable learning dynamics is just not worth it.

---

> ### Author Response · Authors · 2020-11-23
> **The results are not equivalent to linearization**
>
> We thank the reviewer for the appreciation of our work and the critical reading.
>
> **1. The only (minor) issue I have on the presentation side is the name “bidirectional self-normalizing neural networks”.**
> We agree with the reviewer and have changed the name to “bidirectionally self-normalizing neural networks”. Thanks for the correction.
>
> **2. I think that in the end the experimental evaluation demonstrates a major weakness of the proposed approach.**
> Our experiments are indeed on small-scale and artificial. They by design serve the purpose of verifying our theory empirically.
>
> Linear networks would under-fit the datasets. But in our experimental settings, BSNNs with ELU and SELU achieve almost 100% training accuracy, which is not possible for linear networks. The low testing accuracy is due to over-fitting since the networks have huge capacity (200 layers). As trainability and generalization are two separate issues, we only focus on trainability in this paper.
>
> Currently, our method is mostly of theoretical interests. However, as often with foundational work, the experiments are on small-scale. We refer the reviewer to the original paper of LSTM [1] and VAE [2], in which only toy experiments were conducted. It took the community many years to get them work on large-scale applications.
>
> To our knowledge, our work shows for the first time that the vanishing/exploding gradient problem is provably solved for nonlinear neural networks. We hope our theoretical breakthrough can inspire future research in neural network theory and large-scale applications.
>
> [1] Long short-term memory, Hochreiter & Schmidhuber, Neural Computation 1997.
> [2] Auto-encoding variational bayes, Kingma & Welling, ICLR 2014.

---

### Official Review · AnonReviewer2 · 2020-10-27
**Good Work but Need more Analysis**

**Rating:** 6
**Confidence:** 4

**Review:**

Summary:

In this paper, the authors introduce the bidirectional self-normalizing neural networks (BSNN) that preserve the norms in both forward and backward passes. To serve such purpose, a new class of activation functions, GPN, is proposed, which can be obtained via the affine transform of existing activation functions like tanh and SELU. The authors prove that under orthogonal weights and GPN, the norm for both forward and backward passes can be well preserved. Besides, the conclusions are also supported by experiments on synthetic data and real-world data like MNIST and CIFAR 10.


Strength:

--The paper is well organized and easy to follow.

--The proofs seem rigorous.

--The experiments on synthetic data and real-world data are consistent with the theoretical prediction.


Weakness:

--One major strength of the self-normalizing neural network powered by SELU as well as batch normalization is that, even if the statistical properties deviate from the ideal point at some layers, the succeeding layers can gradually fix the deviation. This property greatly stabilizes the training process even under a high learning rate. On the other hand, methods like dedicated initialization and weight normalization/standardization do not have this property, and previous studies have shown that they are less effective compared with SELU and BN. While GPN can be achieved via adding a simple affine transform to the existing activation functions like tanh, does it mean that it is more like an initialization technique? Or it still has the property of fixing deviations in preceding layers?

--For the experiments on real-world data, the depth 200 may be too deep for these small datasets, and the accuracy reported is less satisfying compared with networks with 30-50 layers. Although I understand that the authors want to demonstrate the performance of self-normalization in very deep models, it would be more convincing if the proposed activation functions are also compared with existing ones like SELU under a more practical depth (e.g., 30-50 layers).

---

> ### Author Response · Authors · 2020-11-23
> **Thank you for the appreciation**
>
> We thank the reviewer for the appreciation of our work and the inspiring questions.
>
> **1. While GPN can be achieved via adding a simple affine transform to the existing activation functions like tanh, does it mean that it is more like an initialization technique? Or it still has the property of fixing deviations in preceding layers**
> It is a great question that if the statistical properties or assumptions made in the paper can hold during the training of the networks. We plotted the gradient norm during training in Figure 10-13. It seems the gradients of BSNN are stable throughout training. This might give a hint that the statistical properties are robust.
>
> **2. For the experiments on real-world data, the depth 200 may be too deep for these small datasets, and the accuracy reported is less satisfying compared with networks with 30-50 layers)**
> The vanishing/exploding gradients problem is severe only for deep neural networks. For networks of 30-50 layers, our method does not show significant advantages. We admit the experimental settings seem artificial in that they are not tuned to solve a particular application. They were merely designed to verify our theory empirically given the assumptions we made.

---

### Official Review · AnonReviewer1 · 2020-10-27
**Good idea and theory; substandard experimental part.**

**Rating:** 4
**Confidence:** 4

**Review:**

Summary: The authors introduce a variant
of self-normalizing networks (SNNs) that also maintain
the norm of the errors during backpropagation. The presented
Bidirectional self-normalizing networks (BSNN)
have interesting theoretical properties, but are hardly
assessed empirically. There are severe flaws in
the method evaluation.


Pros:
a) The mathematical formulations are very clear
and sound.
b) The connection between SNNs and works in
mean-field theory (Poole, Schoenholz) are very
well put in place.
c) Making both the forward- and the backward pass
self-normalizing appears desirable for NNs.
d) The work is well embedded into related works.


Cons:
a) Unclear relevance due to lacking and small-scale experiments, furthermore,
due to missing error bars and significance tests.

i) Firstly, SNNs were introduced with relatively large-scale experiments on fully-connected networks.
In order to demonstrate improvements due to bidirectional normalization,
the suggested activation functions should be compared in that set of experiments.
The presented experiments are done with architectures that are extremely far from the current SOTA on CIFAR10,
with 46% versus 90% accuracy already reported in 2013 (see ref[2]). The authors
should perform experiments on large scale, e.g. CIFAR100 and ImageNet or voice recognition [3],
with architectures at or close to the state-of-the-art to demonstrate
the broad applicability and relevance of their method.

ii) It is unclear whether BSNNs represent an advance at all because
all presented performance metrics come without repetitions, error bars
and significance tests. Purported improvements could be just by chance.
The authors should use repetitions, present all metrics with confidence
intervals and perform statistical tests.

iii) Conclusions of the experiments are not justified due to limited
hyperparameter search. The setting of the experiments that activation
functions are compared with the same architecture, same learning rate,
momentum term, and batch size strongly limits the conclusions since
activation functions could work with slightly different settings.
The authors should allow each method (here: activation function)
to optimize hyperparameters.

iv) It is not necessary that almost exact preservation
of the norm in the backward pass must improve learning, especially,
because it is traded against a bias shift.
For standard SNNs, with the SELU activation, there is
a small exploding gradient, which, however leads
to lower layers learning faster [1]. This could even assist learning.
This point should make clear that stringent empirical
assessment is necessary. Therefore, the authors should provide
a much wider empirical assessment of BSNNs.

b) Strong uncommented assumtions. The presented work requires
orthogonal matrices, and several other assumptions. Those assumptions
are left mostly uncommented and leave unclear how applicable this
theory is and how strong it restricts the results and conclusion.
The authors should comment on all assumptions and relate it to learned
neural networks.
i) The assumption of orthogonal weight matrices is a strong one.
This is computationally extremely costly, even in the relaxed way
of the authors (Section 3.2). The authors should elaborate on this.

c) Neglection of learning dynamics. Whereas, the derivation of SNNs
involves learning dynamics by showing that even with changes in the
weights, a fixed points remains, BSNNs neglect learning dynamics.
Even definition of self-normalization requires exact preservations
of the norm. The authors should aim at requiring Eq (4) and (5) to
be kept on expectation and not exactly. Otherwise, no learned
network is self-normalizing.


Minor:


Questions:
a) The expectation over E[\phi(x)] is not more equal zero, which introduces
a bias shift. How does this bias shift interact with learning?
Can you state this difference to SNNs clearer in the manuscript?
b) Lemma 3: Can you comment on how this bound arises from (Ball, 1997)?


Justification of score:
Despite my many points of concerns, the theory in this paper and the main
idea are good. I would be willing
to vote for acceptance if the experimental part is strongly improved.


References:
[1] Hoedt, P.J., Hochreiter, S. and Klambauer, G. Characterising activation functions by their backward dynamics around forward fixed points. Critiquing and Correcting Trends in Machine Learning workshop at NeurIPS 2018.
[2] Goodfellow, I., Warde-Farley, D., Mirza, M., Courville, A., & Bengio, Y. (2013, May). Maxout networks. In International conference on machine learning (pp. 1319-1327). PMLR.
[3] Huang, Z., Ng, T., Liu, L., Mason, H., Zhuang, X., & Liu, D. (2020, May). SNDCNN: Self-normalizing deep CNNs with scaled exponential linear units for speech recognition. In ICASSP 2020-2020 IEEE International Conference on Acoustics, Speech and Signal Processing (ICASSP) (pp. 6854-6858). IEEE.

---

> ### Author Response · Authors · 2020-11-23
> **Focus of the experiments is to verify the theory**
>
> We thank the reviewer for appreciating our theory and the critical reading!
>
> **1. Unclear relevance due to lacking and small-scale experiments, furthermore, due to missing error bars and significance tests.**
> Our experiments were designed with the following question in our mind.
> If the conditions in the theory are mostly satisfied, does the vanishing/exploding gradients problem indeed disappear in practice?
> The experimental settings largely follow from the dynamical isometry paper [1].
>
> It is not our intention to achieve superior performance on standard machine learning benchmarks such as UCI, CIFAR-10 and ImageNet. Although it is a valuable goal on its own, it is separate from our goal.
>
> We thank the reviewer for the experiment suggestions but we found them unsuitable for our theory. For UCI, the neural networks do not need to be deep. In the SNN paper, only 32 layers maximally are needed. The vanishing/exploding gradients problem is not severe for shallow networks. For CIFAR-10 and ImageNet, convolutions and residual connections are employed to achieve superior performance. The two modules are not covered in our theory.
>
> Our experiments are indeed on small-scale and artificial. But they by design serve the purpose of verifying our theory empirically, which is the essence of the experimental part of the paper.
>
> The total running time of the experiments has more than 100 hours, as the networks have 200 layers of 500 units and we have to run each trial for each activation function (with/without batch norm). Given our limited computational resources, we are unable to provide the error bars and exhaustive hyperparameter tuning. We provide full source code allowing researchers to verify our experimental results.
>
> Currently, our method is mostly of theoretical interests. However, as often with foundational work, the experiments are on small-scale. We refer the reviewer to the original paper of LSTM [2] and VAE [3], in which only toy experiments were conducted. It took the community many years to get them work on large-scale applications.
>
> To our knowledge, our work shows for the first time that the vanishing/exploding gradient problem is provably solved for nonlinear neural networks. We hope our theoretical breakthrough can inspire future research in neural network theory and large-scale applications.
>
> [1] Resurrecting the sigmoid in deep learning through dynamical isometry: theory and practice, Pennington, Schoenholz, Ganguli, NIPS 2017.
> [2] Long short-term memory, Hochreiter & Schmidhuber, Neural Computation 1997.
> [3] Auto-encoding variational bayes, Kingma & Welling, ICLR 2014.
>
> **2. Strong uncommented assumtions**
> We are confused by this point. We indeed provided comments in the paper. Please see the paragraph right below the Assumptions “The above assumptions are not restrictive...” on Page 4.
>
> **3. Neglection of learning dynamics**
> Yes, we agree the analysis of learning dynamics is an important topic.  In Figure 10-13 in the Appendix, we show that the gradients are stable for BSNNs throughout training. We leave the mathematical analysis of the learning dynamics to the future work.
>
> **4. The authors should aim at requiring Eq (4) and (5) to be kept on expectation and not exactly.**
> Yes, this is indeed close to our claim. Please see the following paragraph in the paper.
> “Hence, combining Theorems 2 and 3, we proved that bidirectional self-normalization is achievable with high probability if the neural network is wide enough and the conditions in the Assumptions are satisfied. Then by Proposition 1, the gradient exploding/vanishing problem disappears with high probability”
>
> **5. The expectation over E[\phi(x)] is not more equal zero, which introduces a bias shift. How does this bias shift interact with learning?**
> That E[phi(x)] is non-zero  is covered by our theory since E[W*phi(x)] is zero under the uniform distribution of orthogonal W.
>
> **6. Lemma 3: Can you comment on how this bound arises from (Ball, 1997)?**
> Given a vector u on a sphere, the spherical gap is the normalized surface area for vectors v on the sphere with dot(u,v) >= epsilon. Therefore, if random vector v is uniformly distributed on the sphere, the spherical gap is the probability that dot(u,v) >= epsilon. Since P(dot(u,v) >= epsilon) = P(dot(u,v) <= -epsilon) due to symmetry, we have P(|dot(u,v)| <= epsilon) = P(dot(u,v) >= epsilon) + P(dot(u,v) <= -epsilon) and therefore Lemma 3 by applying Lemma 2.2 in (Ball, 1997).
>
> The result is known as “near orthogonality” for two uniformly distributed random unit vectors. We refer the reviewer to Lemma B.1 in [3] for a similar proof.
> [3] The Impact of Neural Network Overparameterization on Gradient Confusion and Stochastic Gradient Descent, Sankararaman et al., ICML 2020.
>
> We simplified the proof of the main results in the revised paper. The proof is more readable than the previous one. The Lemma 3 in the previous version is no longer needed.

---

### Official Review · AnonReviewer3 · 2020-10-28
**A completion of sorts for an interesting line of work**

**Rating:** 6
**Confidence:** 3

**Review:**

This paper presents an interesting approach to normalizing the signals in a deep network both in the forward pass and in the backward pass with the goal of preventing vanishing and exploding gradients for better training convergence (and perhaps general elegance).

Previously, this has been partly solved by the use of orthogonal weight matrices (and this completely solves the problem for linear networks). The authors show that additionally a specific modification of the non-linearity (essentially a scale and shift) is enough to guarantee good bi-directional normalization with high probability.

I did not check the math. It would be good if the authors could provide some intuition for the main contribution: that an appropriate scaling and shifting of the nonlinearity is what is needed.

Empirical results are demonstrated in essentially toy settings (synthetic, and mnist/cifar with fully connected networks) but these experiments are thorough (but see question below).

My positive review of the paper is mostly based on viewing this as a natural completion of an existing line of work (around the use of orthogonal weight matrices for normalization), but I worry that this line of work may not lead to practically useful results since the orthogonality constraint is hard to enforce exactly, and it is not clear if this would work for convolution given the requirement for sufficient width. This is why I am hesitant to give it a higher score.

Questions, mostly around the practicality of the proposed techniques.

1. How would this work on convolutional networks where one does not typically expect large “width” i.e. is this work only of theoretical interest?

2. How does this impact generalization? There is reason to believe that mucking with gradients can hurt generalizations. This has been discussed for  adaptive methods (Duchi et al., etc. alluded to the in the introduction) in https://arxiv.org/abs/1705.08292.

3. How would you compare with residual networks? Particularly in terms of run-time and generalization.

Minor:

a. What is the significance of Theorem 1?

b. Theorem 2 statement should say $\phi$ is Gaussian-Poincare normalized?

---

> ### Author Response · Authors · 2020-11-23
> **Thank you for the appreciation**
>
> We thank the reviewer for the appreciation of our work and the inspiring questions!
>
> **1. It would be good if the authors could provide some intuition for the main contribution**
> We provided some intuition of our main contribution in the “Sketch of the proofs” paragraph. The shift and scaling of activation allow us to apply the concentration of norm property (Lemma 2). Here, we give the following Python snippet. Running the code will provide some intuition into the theory.
>
> ~~~
> import torch
>
> # Forward norm-preservation
> z = torch.randn(10000)
> f = 1.4674 * torch.tanh(z) + 0.3885
> print(z.norm(), f.norm())
>
> # Backward norm-preservation
> z = torch.randn(10000)
> f = 1.4674 * (1-torch.tanh(z)**2)
> x = torch.rand(10000)
> y = f * x
> print(x.norm(), y.norm())
> ~~~
>
> **2. How would this work on convolutional networks where one does not typically expect large “width” i.e. is this work only of theoretical interest?**
> Currently, our theory is developed only for fully connected networks. Additionally, large width convolutional networks have achieved some empirical successes [1]. In the future work, we plan to extend our theory to more sophisticated architectures such as convolutional networks, recurrent networks and networks with skip connections (e.g., ResNet).
>
> [1] Wide Residual Networks,
> Zagoruyko & Komodakis, BMVC 2016.
>
> **3. How does this impact generalization?**
> Understanding the relationship between gradient behaviors and generalization is certainly an interesting topic but however beyond the scope of this paper. There are claims that trainability and generalization are separate problems [2]. We only investigate trainability in this paper. We look forward to seeing more work done on this topic by ourselves and the community in the future.
>
> [2] Disentangling Trainability and Generalization in Deep Neural Networks,
> Xiao, Pennington, Schoenholz, ICML 2020.
>
> **4. How would you compare with residual networks?**
> In our experimental settings, adding residual connections often leads to gradient explosion (with or without BatchNorm). Currently, our theory is developed for fully connected networks and incompatible with residual networks. The reason is that orthogonal matrices and GPN functions are norm-preserving (under the conditions given in the paper) but adding residual connection is not. In the future, we plan to extend our theory to residual networks.
>
> **5. What is the significance of Theorem 1?**
> Theorem 1 shows the fundamental relationship between a function and its derivative under Gaussian measure. It inspires us to come up with the Gaussian-Poincare Normalization, which has a similar flavor. Additionally, the proof of Proposition 3 relies on Theorem 1.
>
> **6. Theorem 2 statement should say Phi(x) is Gaussian-Poincare normalized?**
> Yes, both Theorem 2 and 3 rely on the Assumptions, as we stated in the paper.

---

### Decision · Program_Chairs · 2021-01-07
**Final Decision**

**Decision:**

Reject

**Comment:**

Three knowledgeable referees rate this paper ok but not good enough or borderline positive (4,4,6), and one fairly confident referee rates it borderline positive 6. The referees discussed the authors' responses and, while they considered the idea and some of the theory good, they remain concerned, in particular about the experimental part and generalization discussion. The scores remained unchanged after the discussion. Hence I must reject the article.